# FairLISA: Fair User Modeling with Limited Sensitive Attributes Information

**Zheng Zhang**[1,2]    **Qi Liu**[1,2*]    **Hao Jiang**[1,2]    **Fei Wang**[1,2]    **Yan Zhuang**[1,2]
**Le Wu**[3]    **Weibo Gao**[1,2]    **Enhong Chen**[1,2]
1: Anhui Province Key Laboratory of Big Data Analysis and Application
University of Science and Technology of China
2: State Key Laboratory of Cognitive Intelligence
3: Hefei University of Technology
{zhangzheng,jianghao0728,wf314159,zykb,weibogao}@mail.ustc.edu.cn;
{qiliuql,cheneh}@ustc.edu.cn;
lewu.ustc@gmail.com

## Abstract

User modeling techniques profile users' latent characteristics (e.g., preference) from their observed behaviors, and play a crucial role in decision-making. Unfortunately, traditional user models may unconsciously capture biases related to sensitive attributes (e.g., gender) from behavior data, even when this sensitive information is not explicitly provided. This can lead to unfair issues and discrimination against certain groups based on these sensitive attributes. Recent studies have been proposed to improve fairness by explicitly decorrelating user modeling results and sensitive attributes. However, most existing approaches assume that fully sensitive attribute labels are available in the training set, which is unrealistic due to collection limitations like privacy concerns, and hence bear the limitation of performance. In this paper, we focus on a practical situation with limited sensitive data and propose a novel FairLISA framework, which can efficiently utilize data with known and unknown sensitive attributes to facilitate fair model training. We first propose a novel theoretical perspective to build the relationship between data with both known and unknown sensitive attributes with the fairness objective. Then, based on this, we provide a general adversarial framework to effectively leverage the whole user data for fair user modeling. We conduct experiments on representative user modeling tasks including recommender system and cognitive diagnosis. The results demonstrate that our FairLISA can effectively improve fairness while retaining high accuracy in scenarios with different ratios of missing sensitive attributes.

## 1 Introduction

User modeling is a fundamental task in many applications (e.g., recommender system), which aims to infer latent characteristics through analyzing users' behavioral information, such as capability and preference [56]. The results from user modeling serve as an important basis for decision-making, which might affect users' lives to some extent [17, 15]. However, recent studies have demonstrated that biases related to sensitive attributes (e.g., gender, race) can be unconsciously captured from behavior data, even when this sensitive information is not explicitly provided. This can lead to unfair user modeling results [3, 23, 45]. For example, college admissions based on user modeling may underestimate underrepresented demographic groups [48]. Career recommendation shows apparent gender discrimination even for equally qualified men and women [19]. As such, it is crucial to consider fairness issues and ensure that users with different sensitive attributes are treated similarly.

---

*Corresponding Author

37th Conference on Neural Information Processing Systems (NeurIPS 2023).

Following previous studies [3, 23, 45], the fairness requirement for user modeling in this paper is that the results should not expose any sensitive information. This requirement can be formalized as minimizing the mutual information between user modeling results and sensitive attributes [2]. Along this way, many approaches have been proposed, such as constraint optimization [1], adversarial learning [3, 23, 45, 20] and regularization methods [49, 42], among which adversarial methods show their theoretical elegance and receive widespread attention.

While many methods have shown promising results, most of them require explicit sensitive attributes to guide fair model training. However, in real-world scenarios, users are not always willing to share sensitive information due to privacy concerns. For example, only 14% of teen users expose their complete profiles on Meta [31]. The limited sensitive attributes situation means that only a limited amount of data with known sensitive attributes can participate in model training, resulting in insufficient training data problem for existing methods. The most practical solution is to utilize data with unknown sensitive attributes to facilitate model training. Nevertheless, several obstacles must be addressed, including: 1) **Efficient data utilization**: Currently, some researchers have utilized data with unknown sensitive attributes by predicting missing sensitive labels based on data with known sensitive attributes [5]. However, the accuracy of these predicted labels cannot be guaranteed, especially in extremely limited sensitive situations, which results in suboptimal data utilization. 2) **Theoretical guarantee**: The fair goal in our paper is to minimize the mutual information between the sensitive attributes and the user modeling results. When we utilize data with unknown sensitive attributes to improve fair user modeling, how can we establish a good theoretical guarantee? 3) **Framework generalization**: Several fair adversarial methods have been introduced to handle data with known sensitive attributes [3, 23, 45, 20]. Each method has its advantages and disadvantages in different scenarios. In this case, we believe that a general framework should be proposed that can incorporate data with unknown sensitive attributes into existing fair adversarial methods.

To address these obstacles, in this paper, we propose FairLISA, a general framework for *Fair user modeling with LImited Sensitive Attributes*, which can efficiently utilize data with known and unknown sensitive attributes to facilitate fair model training. We first propose a novel theoretical perspective to build the relationship between two types of data and the fair mutual information objective. Specifically, we establish a connection between data with unknown sensitive attributes and the fairness goal through shannon entropy, while data with known sensitive attributes and the fairness goal are connected through cross-entropy. By doing so, we can directly leverage user data with unknown sensitive attributes without the need for predicting missing attributes, reducing information loss caused by predictions and ultimately increasing the utilization rate of data with unknown sensitive attributes. Then building on this foundation, we propose an adversarial learning approach to effectively leverage the entire user data for fair model training. Since existing fair adversarial approaches requiring fully known sensitive data can be interpreted from a mutual information perspective [3, 23, 45], FairLISA can easily generalize most existing adversarial methods and expand them to limited sensitive attribute situations. Finally, to validate the effectiveness of our proposed FairLISA framework, we conduct extensive experiments on two representative user modeling tasks including the recommender system and cognitive diagnosis. Empirical results show that our framework can enhance fairness with satisfactory accuracy performance, even in scenarios with different ratios of missing sensitive attributes. The main contributions of this work are as follows:

- *Efficient Data Utilization and Theoretical Guarantee.* We provide a novel theoretical perspective to build the relationship between data with known and unknown sensitive attributes with fairness objective. This allows for the effective leveraging of data with unknown sensitive attributes, reducing information loss caused by predicting missing attributes.

- *Framework Design and Generalization.* We propose a general framework, FairLISA, which can expand most existing fair adversarial methods to limited sensitive situations.

- *Experimental Evaluations.* We conduct experiments in two representative user modeling tasks (i.e., recommender system, cognitive diagnosis) with six representative models and two datasets, the results demonstrate the effectiveness and robustness of FairLISA.

## 2 Related work

**User Modeling.** User modeling is a fundamental task that captures users' useful potential characteristics, such as capability, preference, and so on [44, 56, 28, 50]. It has been applied in various

applications. For example, based on user capability fitting, researchers employed cognitive diagnosis [30, 9] to model student proficiency [4, 30]; based on user preference mining, researchers applied collaborative filtering [39] for mining user interest preferences to different tasks including news recommendation [26], social network [46]. Most of the existing user modeling methods focused on getting more precise results [4, 30, 16, 15, 27, 29]. However, as the results directly affect the opportunity of users [32, 54, 55], it can easily lead to unfair outcomes. In this paper, we explore fairness-aware user modeling, which aims to ensure fairness during the user modeling process.

**Fairness in Machine Learning.** Fairness in machine learning can be divided into three categories: (1) individual fairness, which requires similar individuals to be treated similarly [7, 6]; (2) group fairness, which requires different groups divided by sensitive attributes should be treated similarly [35, 7, 13]; (3) max-min fairness, which aims to maximize the utility of worst-off group [37]. In this work, we focus on group fairness since it can measure how the underrepresented group is treated.

In group fairness research, many fairness metrics and unfairness mitigation techniques had been proposed [22, 36, 45, 38, 47, 3, 23, 43, 10, 21]. For example, Avishek et al. [3] discovered the learned user representations would capture biases of sensitive information and proposed to minimize the mutual information between representations and sensitive attributes through an adversarial framework. Li et al. [23] also proposed this goal from the causal perspective and provided two similar adversarial strategies. Wu et al. [45] argued the user-centric structure would expose users' sensitive attributes and proposed an adversarial framework from a graph perspective to obtain more efficient results.

Despite their ability to improve fairness, all of these methods are based on enough user sensitive attribute information. While in many real-world applications, it is difficult to collect such data since users may care about privacy and are reluctant to divulge their sensitive information [31]. The lacking of sensitive information challenges most existing solutions. Exploring fair models with missing sensitive attributes is an important and challenging issue, and it is still under-explored. There are only a few works in this direction. One branch of approaches investigates fairness without sensitive attributes. For example, some researchers achieved this via solving a max-min problem [25, 14, 18]. However, they were only effective for max-min fairness [37], which was different from our group fairness goal. Some researchers tried to solve this problem with the help of related features [51, 53]. However, obtaining such related features is not always feasible in practical applications, thereby limiting the utilization of these methods. The other branch explores improving fairness in limited sensitive attribute situations, Dai et al. [5] proposed FairGNN, the main idea was to predict missing sensitive attribute labels. However, their reliance on the accuracy of the predicted pseudo labels resulted in suboptimal data utilization, particularly in extremely limited sensitive situations. Different from previous papers, we explore group fairness in limited sensitive attribute situations and can efficiently utilize data with known sensitive attributes without predicting missing sensitive attributes.

## 3 Problem Definition

In user modeling, we denote $U = \{u_i\}_{i=1}^{N_u}$ and $V = \{v_i\}_{i=1}^{N_v}$ as the user set and the item set, respectively. The interactions between users and items generate interaction records $R$, whose element is denoted as triplet $(u, v, r)$ where $u \in U$, $v \in V$, and $r$ is the corresponding interaction decided by the corresponding task. As we have assumed that sensitive attributes are only available for a limited number of users, we split the user set $U$ into two disjoint sets as $U_L = \{u_i\}_{i=1}^{l}$ and $U_N = \{u_i\}_{i=l+1}^{l+n}$ so that the sensitive label $s$ is only accessible for each user in $U_L$ instead of $U_N$. Let $S$ be the sensitive attribute that takes value in $\mathcal{S}$, whose values for users in $U_L$ are available, i.e. $\{s_i|s_i \in \mathcal{S}\}_{i=1}^{l}$. Our goal is to learn a user model $f$ that generates a fair representative characteristic $\theta$ for each user $u$. Following previous studies [3, 21, 42], the fairness requirement for user modeling is that the generated outcome should not expose any sensitive information, which can be depicted as the result of zero mutual information between the user's generated characteristic $\theta$ and sensitive attribute $S$, i.e. $I(\theta, S) = 0$. In this way, we have the following problem definition:

**Problem 1.** *Given interaction records $R$, users $U$, items $V$ and the sensitive label set $\{s_i\}_{i=1}^{l}$ for $U_L \subset U$, the goal is to learn a user model $f : U \to \Theta$ that generates a **fair** representative characteristic $\theta \in \Theta$ for each user $u \in U$, such that $I(\theta, S) = 0$ where $S$ is the corresponding sensitive attribute.*

# 4 FairLISA

## 4.1 Overview

FairLISA is an adversarial learning-based framework designed to promote fair model training by efficiently utilizing data with known and unknown sensitive attributes. The architecture is shown in Figure 1. It consists of two modules including a filter $\mathcal{F}$ and a discriminator $\mathcal{D}$. Specifically, for a given user with the original representation $\theta$ generated by any user models, we use the filter $\mathcal{F}$ trained on data with known sensitive attribute $U_L$ and data with unknown sensitive attribute $U_N$ to filter out the information of sensitive attributes. The filtered user modeling result of a user $u$ is denoted as $\hat{\theta} = \mathcal{F}(\theta)$. To guide the filter $\mathcal{F}$ training, we use the idea of adversary learning to train a discriminator $\mathcal{D}$. The goal of the discriminator $\mathcal{D}$ is to fail the filter $\mathcal{F}$, which is achieved by predicting the corresponding sensitive attribute from the filtered user representations $\hat{\theta}$. Through adversarial training between $\mathcal{F}$ and $\mathcal{D}$, we can get a filter to remove the effect of sensitive attributes. Unlike traditional adversarial methods [3, 23, 45] that only rely on $U_L$, FairLISA can train a filter

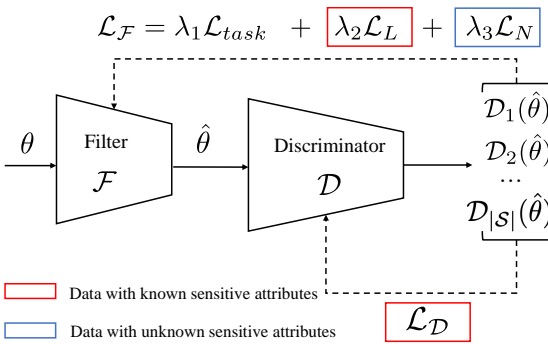

Figure 1: The architecture of FairLISA. For a given user with the original representation $\theta$ generated by any user models, we first use the filer $\mathcal{F}$ trained on $U_L$ and $U_N$ to remove the sensitive information and get the filtered representation $\hat{\theta}$. Then we use the discriminator $\mathcal{D}$ trained on $U_L$ to predict the sensitive feature of $\hat{\theta}$. Through adversarial training and the better utilization of $U_N$, we can get a better filter to remove sensitive information.

with $U_L$ and $U_N$, i.e., $\hat{\theta} = \mathcal{F}_{U_L+U_N}(\theta)$. Following the previous study [5], we assume the distribution of users' filtered representations in $U_L$ and $U_N$ to be the same and use $p(\hat{\theta})$ to represent them. We will introduce how FairLISA achieves the fairness goal (i.e., $I(\hat{\theta}, S) = 0$) through $U_L, U_N$ respectively and how to utilize these two types of data to form the final architecture. For simplicity of theoretical analysis, we assume there is one sensitive attribute $S$ (e.g., gender) and $s$ (e.g., female) is an attribute value of $S$.

## 4.2 Fairness with Known Sensitive Attributes

We will begin by explaining how FairLISA achieves $I(\hat{\theta}, S) = 0$ with data with known sensitive attributes $U_L$, i.e., $\hat{\theta} = \mathcal{F}_{U_L}(\theta)$. More specifically, we will describe how to establish a connection between $U_L$ and $I(\hat{\theta}, S)$, as well as how to leverage data with known sensitive attributes to minimize $I(\hat{\theta}, S)$ based on this connection. In this regard, we define $\mathcal{D}_s(\hat{\theta})$ as the probability that the discriminator $\mathcal{D}$ recognizes the value of the sensitive attribute $S$ as $s$ based on $\hat{\theta}$, $H$ denotes the shannon entropy [24]. According to the following lemma, mutual information $I(\hat{\theta}, S)$ can be expressed as the optimization function shown below, when the optimal discriminator can achieve $\mathcal{D}_s^*(\hat{\theta}) = p(s|\hat{\theta})$.

**Lemma 4.1.** *Given the optimal discriminator $\mathcal{D}_s^*(\hat{\theta})$ and let $p(\hat{\theta}|s)$ be the distribution of users' filtered representations whose sensitive attribute value is $s$, then $I(\hat{\theta}, S)$ can be derived as,*

$$I(\hat{\theta}, S) = \sum_s p(s)\mathbb{E}_{\hat{\theta}\sim p(\hat{\theta}|s)}\left[\log \mathcal{D}_s^*(\hat{\theta})\right] + H(S). \tag{1}$$

The detailed proof is provided in Appendix A.1. We have established a connection between $U_L$ and $I(\hat{\theta}, S)$ through Lemma A.1. Consequently, fairness can be achieved by minimizing Eq. (11) with a fixed optimal discriminator $\mathcal{D}^*$. It is important to note that estimating $p(\hat{\theta}|s)$ in Eq. (11) requires access to sensitive information. This implies that leveraging $U_N$ (data with unknown sensitive attributes) to enhance fairness becomes impossible. However, in real-world applications, users are often concerned about privacy and may be unwilling to disclose their sensitive information. The limited availability of sensitive attributes means that only a restricted amount of data with known

sensitive attributes can participate in model training, leading to an insufficient training data problem for existing methods. In the following sections, we will introduce how to alleviate this problem by effectively utilizing data with unknown sensitive attributes.

### 4.3 Fairness with Unknown Sensitive Attributes

An intuitive approach to leverage data with unknown sensitive attributes $U_N$ is to predict the missing sensitive labels based on $U_L$. However, the accuracy of these predicted pseudo-labels cannot be guaranteed, which results in suboptimal data utilization. Therefore, it is crucial to explore new methods that fully harness the potential of $U_L$ to enhance fairness. Given that the suboptimal data utilization stems from predicting missing attributes, a natural idea is to directly employ $U_N$ without predicting missing attributes, which can avoid information loss caused by prediction and maximize the utilization of $U_N$ to a greater extent. To achieve this objective, it is necessary to directly establish a relationship between $I(\hat{\theta}, S)$ and $U_N$. Drawing inspiration from Lemma A.1 and structural insights related to the KL divergence [11], we conclude that shannon entropy $H$ can build this relationship.

**Lemma 4.2.** *Given the optimal discriminator $\mathcal{D}_s^*(\hat{\theta})$ and let $p(\hat{\theta})$ be the distribution of users' filtered representation, we have,*

$$
\begin{aligned}
I(\hat{\theta}, S) &= \mathbb{E}_{\hat{\theta} \sim p(\hat{\theta})} \sum_s \left[ \mathcal{D}_s^*(\hat{\theta}) \log \mathcal{D}_s^*(\hat{\theta}) \right] + H(S) \\
&= -H(\mathcal{D}_s^*(\hat{\theta})) + H(S).
\end{aligned}
\tag{2}
$$

*Proof.* According to the relationship between mutual information and JS divergence, the relationship between JS divergence and KL divergence [24], we have

$$
I(\hat{\theta}, S) = JS_{p(s)}(p(\hat{\theta}|s)) = \sum_s p(s) KL(p(\hat{\theta}|s) \| p(\hat{\theta})).
\tag{3}
$$

By calculating KL divergence, we get,

$$
\sum_s p(s) KL(p(\hat{\theta}|s) \| p(\hat{\theta})) = \int \sum_s p(s) p(\hat{\theta}|s) \log \frac{p(\hat{\theta}|s)}{p(\hat{\theta})} d\hat{\theta}.
\tag{4}
$$

Then we expand the formula and use bayes' theorem,

$$
\begin{aligned}
&\int \sum_s p(s) p(\hat{\theta}|s) \log \frac{p(\hat{\theta}|s)}{p(\hat{\theta})} d\hat{\theta} \\
&= \int p(\hat{\theta}) \sum_s p(s|\hat{\theta}) \log p(s|\hat{\theta}) d\hat{\theta} + H(S).
\end{aligned}
\tag{5}
$$

Noted that $p(s|\hat{\theta})$ is exactly $\mathcal{D}^*$, we replace it in the formula, and get the result,

$$
\begin{aligned}
&\int p(\hat{\theta}) \sum_s p(s|\hat{\theta}) \log p(s|\hat{\theta}) d\hat{\theta} + H(S) \\
&= -H(\mathcal{D}^*(\hat{\theta})) + H(S).
\end{aligned}
\tag{6}
$$

$\square$

Here we give the key steps of proof, more details are provided in Appendix A.2. In Lemma A.2, the term $p(\hat{\theta}|s)$ disappears, removing the obstacle that hindered the direct utilization of $U_N$ as discussed in Lemma A.1. Thus, we have successfully established the relationship between $I(\hat{\theta}, S)$ and $U_N$ through Lemma A.2. We also provide an intuitive illustration. By maximizing the entropy of the discriminator, we can ensure that the prediction probability of sensitive attributes becomes uniform. For example, in the case of gender prediction, both male and female probabilities are set to 0.5. This shows that the discriminator becomes unable to predict the sensitive features from the learned user embedding. Consequently, we can minimize our fairness criterion $I(\theta, S)$ directly by minimizing Eq. (15) using $U_N$. This allows for the effective utilization of $U_N$ to enhance fairness without the need to predict unknown sensitive attribute labels.

## 4.4 Architecture

In this section, we now introduce how to combine $U_L$ and $U_N$ together to form the final architecture (shown in Figure 1) in order to obtain a higher improvement of fairness, i.e., $\hat{\theta} = \mathcal{F}_{U_L+U_N}(\theta)$. By leveraging the insights from Lemma A.1 and Lemma A.2, we have established the connection between $U_L$, $U_N$, and the fairness goal $I(\hat{\theta}, S)$. Thus, fairness can be achieved through minimizing Eq. (11) with $U_L$ and minimizing Eq. (15) with $U_N$ with a fixed optimal discriminator $\mathcal{D}^*$. To achieve this goal, we first optimize the $\mathcal{D}$ to the best, then fix $\mathcal{D}$ to optimize $\mathcal{F}$. FairLISA consists of a filter and a discriminator. We will introduce them respectively as well as the training algorithm.

**Filter.** We use the whole data to optimize the $\mathcal{F}$. For data with known sensitive attributes, we optimize $\mathcal{F}$ by minimizing the right side of Eq. (11). The empirical estimate can be computed as:

$$\mathcal{L}_L = \frac{1}{l} \sum_{s} \sum_{1 \leq i \leq l : s_i = s} \log \mathcal{D}_s(\hat{\theta}_i), \tag{7}$$

where $\hat{\theta}_i$ is the filtered representation of $u_i$ . Here we omit $H(S)$ in Eq. (11) since it is a constant.

For data with unknown sensitive attributes, we optimize $\mathcal{F}$ by minimizing the right side of Eq. (15). The empirical estimate can be computed as:

$$\mathcal{L}_N = \frac{1}{n} \sum_{l+1 \leq i \leq l+n} \sum_{s} \mathcal{D}_s(\hat{\theta}_i) \log(\mathcal{D}_s(\hat{\theta}_i)), \tag{8}$$

where $\hat{\theta}_i$ is the filtered representation of $u_i$ . Here we also omit $H(S)$ since it is a constant.

Furthermore, in addition to the fairness requirement, we need to maintain users' useful potential characteristics for different user modeling tasks (e.g., recommender system, cognitive diagnosis). To achieve this, we use a task-specific loss function to mine useful information of users. With the combination of Eq. (7) and Eq. (8), we get the following loss function:

$$\min_{\mathcal{F}} \mathcal{L}_{\mathcal{F}} = \lambda_1 \mathcal{L}_{task} + \lambda_2 \mathcal{L}_L + \lambda_3 \mathcal{L}_N, \tag{9}$$

where $\mathcal{L}_{task}$ represents the loss of specific user modeling task, $\lambda_1$, $\lambda_2$ and $\lambda_3$ are hyperparameters. We study the influence of these hyperparameters in the experiment section (RQ3).

**Discriminator.** To guide the filter $\mathcal{F}$ training, the goal of the discriminators is to predict the corresponding sensitive attribute from the filtered user representations $\hat{\theta}$. Therefore, for the loss function of $\mathcal{D}$, we optimize the discriminator with data $U_L$ using the following cross-entropy:

$$\min_{\mathcal{D}} \mathcal{L}_{\mathcal{D}} = -\frac{1}{l} \sum_{s} \sum_{1 \leq i \leq l : s_i = s} \log \mathcal{D}_s(\hat{\theta}_i), \tag{10}$$

where $\hat{\theta}_i$ is the filtered representation of $u_i$ . Here we omit $H(S)$ in Eq. (11) since it is a constant..

**Training Algorithm.** Since the discriminator can not be optimized to the best directly, following the traditional adversary learning training methods [12], we adopt mini-batch training in implementation. Specifically, 1) $T$ mini-batch updates minimizing $\mathcal{L}_F$ with the $\mathcal{D}$ fixed; 2) $T$ mini-batch updates minimizing $\mathcal{L}_D$ with the $\mathcal{F}$ fixed. Here $T = 10$ in our implementation. The whole optimization algorithm for FairLISA is shown in Appendix B.

## 4.5 Relation to Existing Adversarial Methods

In this paper, we provide a novel mutual information perspective to interpret the strategies for achieving fairness in existing adversarial methods, and unify the treatment of data with known and unknown sensitive attributes. As a result, FairLISA can easily generalize most existing fairness-aware adversarial methods with data with unknown sensitive attributes. For example, if we set $\lambda_3$ in Eq. (9) to 0, which means ignoring the data in $U_N$, FairLISA degenerates to ComFair [3], a classical fairness-aware adversarial method. Moreover, our fairness mutual information objective is compatible with existing adversarial methods and can theoretically expand them to the limited sensitive attribute situation. For example, if we combine Eq. (8) from FairLISA with FairGo [45], a state-of-the-art fair model which considers graph structure, we could enhance its fairness performance. We will show the performance in experiments (denoted as FairGo+FairLISA).

# 5   Experiments

In this section, we first introduce the dataset and experimental setup. Then, we conduct extensive experiments to answer the following questions:

- **RQ1**: Does FairLISA outperform the fairness-aware baselines in limited sensitive situations?

- **RQ2**: Can FairLISA achieve more robust fairness results on different missing ratios?

- **RQ3**: How do the data with known and unknown sensitive attributes impact the results?

- **RQ4**: How is the performance of FairLISA on several classical group fairness metrics?

The code is released at https://github.com/zhengz99/FairLISA.

## 5.1   User Modeling Tasks

To explore fairness in user modeling with limited sensitive information, we choose two representative issues in real-world user modeling scenarios. The first is user capability modeling in areas such as education which is called cognitive diagnosis [40]. The user $u$, item $v$ in our general user modeling refer to student and exercise correspondingly, and interaction result $r$ is the score that the student got on exercise. The goal is to model the capability of students. To achieve this, we train models on student performance prediction task. We choose several representative models: IRT [30], MIRT [4], NCD [41]. The second is user preference modeling in recommender system [39]. The user $u$, item $v$ here refer to the customer and product respectively. The interaction result in $r$ is the user evaluation of the product, such as rating behavior. By user rating prediction task, we can mine user preference. We choose several representative models: PMF [34], NCF [15], LightGCN [16]. Detailed basic user model descriptions will be shown in Appendix C.1.

## 5.2   Dataset Description

We use two real-world datasets to verify the effectiveness and robustness of FairLISA. Specifically, we use PISA2015[2] for cognitive diagnosis and MovieLens-1M[3] for a recommender system. Detailed descriptions can be found in Appendix C.2. Both of these datasets are publicly available and do not contain any personally identifiable information. In our experiments, we split each dataset into training, validation, and testing sets, with a ratio of 7:1:2. All baseline models were trained and evaluated using the same datasets.

## 5.3   Experimental Setup and Baselines

**Evaluation.**   The evaluation of our methods can be divided into accuracy evaluation and fairness evaluation. For accuracy evaluation, we select different accuracy metrics for different tasks. For cognitive diagnosis, following previous works from cognitive diagnosis [41, 8], we adopt different metrics from the perspectives of both regression (MAE, RMSE) and classification (AUC, ACC). For recommender system task, we adopt RMSE.

For fairness evaluation, our fairness goal refers to user modeling results do not leak any user's sensitive information. To measure the fairness performance, following the settings in fair user modeling such as [3, 23, 45], we train an attacker who has the same structure as the discriminator. Specifically, after training our framework, we use the filtered user representations as input and the corresponding sensitive attributes as labels to train the attacker. If the attacker can distinguish sensitive features from the user representations, it indicates that the sensitive information is exposed by user modeling results. To train and evaluate attackers, we split the data into train (80%) and test sets (20%) and calculate the AUC score and apply their macro-average to make the result insensitive to imbalanced data. The smaller values of AUC denote better fairness performance with less sensitive information leakage, which be used as the unfairness metric in the following experiments.

---

[2]https://www.oecd.org/pisa/data/2015database/
[3]https://grouplens.org/datasets/movielens/1m/

Table 1: The fairness and accuracy performance on recommender system task. AUC represents AUC scores of all attackers. The smaller values of AUC denote better fairness performance with less sensitive information leakage (the fairer). G, A, O represent gender, age, and occupation. RMSE represents accuracy performance. The best fairness and accuracy results methods are highlighted in bold. The runner-up accuracy results are represented by underline.

| | PMF | | | | NCF | | | | LightGCN | | | |
|---|---|---|---|---|---|---|---|---|---|---|---|---|
| | AUC-G | AUC-A | AUC-O | RMSE | AUC-G | AUC-A | AUC-O | RMSE | AUC-G | AUC-A | AUC-O | RMSE |
| Origin | 0.6862 | 0.7235 | 0.6656 | **0.8670** | 0.6915 | 0.7153 | 0.6625 | **0.8635** | 0.7112 | 0.7053 | 0.6939 | **0.8301** |
| ComFair | 0.5389 | 0.5554 | 0.5398 | 0.9027 | 0.5379 | 0.5414 | 0.5374 | 0.8994 | 0.5498 | 0.5498 | 0.5411 | 0.8704 |
| FairGo | 0.5592 | 0.5592 | 0.5371 | 0.8933 | 0.5351 | 0.5412 | 0.5319 | 0.8941 | 0.5457 | 0.5435 | 0.5408 | 0.8658 |
| FairGNN | 0.5288 | 0.5370 | 0.5212 | 0.9025 | 0.5294 | 0.5305 | 0.5256 | 0.9009 | 0.5289 | 0.5315 | 0.5225 | 0.8648 |
| FairLISA | 0.5193 | **0.5240** | 0.5210 | 0.8935 | 0.5281 | 0.5220 | 0.5222 | 0.8853 | 0.5273 | 0.5291 | 0.5214 | 0.8564 |
| FairGo+FairLISA | **0.5132** | 0.5270 | **0.5112** | 0.8927 | **0.5273** | **0.5204** | **0.5214** | 0.8867 | **0.5241** | **0.5260** | **0.5208** | 0.8628 |

**Implementation detail.** For task-specific loss $\mathcal{L}_{task}$ in Eq.(9), we use cross-entropy loss for cognitive diagnosis tasks and MSE loss for recommendation tasks. For all datasets and models, we set the learning rate to 0.001 and the dropout rate to 0.2. We select the best model based on its original performance on the valid set and use it to conduct the fairness test. We set adversarial coefficient $\lambda_1 = 1$, $\lambda_2 = 2$, $\lambda_3 = 1$ for cognitive diagnosis, and $\lambda_1 = 1$, $\lambda_2 = 20$, $\lambda_3 = 10$ for recommendation tasks. For the adversarial architecture, the filter module is two-layer perceptrons and we use ReLU as the activation function. The discriminators and attackers are three-layer perceptrons with the activation function of LeakyReLU. We set the dropout rate to 0.1 and the slope of the negative section for LeakyReLU to 0.2 for them. We implement all models with PyTorch and conduct all experiments on four 2.0GHz Intel Xeon E5-2620 CPUs and a Tesla K20m GPU.

**Baseline approaches.** To validate the effectiveness of FairLISA, we compare FairLISA with the following baselines. Origin (i.e., basic user models without fairness consideration), ComFair [3], FairGo [45], FairGNN [5]. Further, to validate the generalizability of FairLISA, we expand FairGo to FairGo+FairLISA (introduced in section 4.5). A detailed description will be shown in Appendix C.3.

## 5.4 Experimental Results

**Overall results (RQ1).** The fairness and accuracy results of the recommendation task are shown in Table 1. The results of cognitive diagnosis are shown in Appendix C.4. We have several observations.

- From the perspective of model fairness, the AUC of attackers on origin user models is significantly higher than 0.5, which means that the sensitive information is exposed by user modeling results even if such information is not explicitly used. Further, models which can simultaneously apply data with known and unknown sensitive information (i.e., FairGNN, FairLISA, FairGo+FairLISA) get better fairness performance. It shows that data with unknown sensitive attributes benefit fair user modeling. More importantly, we find our methods (i.e., FairLISA, FairGo+FairLISA) outperform others, suggesting the effectiveness of our work.

- From the perspective of model accuracy, all fairness-improving methods lose some accuracy compared with origin user models. The reason is that the fairness-aware methods are aiming at filtering out the information of certain sensitive features from user modeling results, which will to some extent reduce the information contained, thus decreasing accuracy [3, 33, 45]. Among these fairness-aware methods, we could find FairLISA, FairGo+FairLISA have the best accuracy performance, and they could even maintain comparable accuracy with origin models in some cases.

**Performance on different ratios of missing sensitive attributes (RQ2).** In this part, we study the impacts of sensitive attributes missing ratios and investigate the robustness of FairLISA in extremely limited sensitive situations. We choose the representative models which achieve the best performance result in RQ1 (i.e., NCD, LightGCN) and vary the ratio of missing sensitive attributes as {20%, 40%, 60%, 80%, 95%} in the training set. From the results in Figure 2, we have the following findings.

- The sensitive attributes missing ratios have a huge impact on the fairness results. As the missing ratios increase, the fairness-aware methods are not effective enough. Especially, when the missing

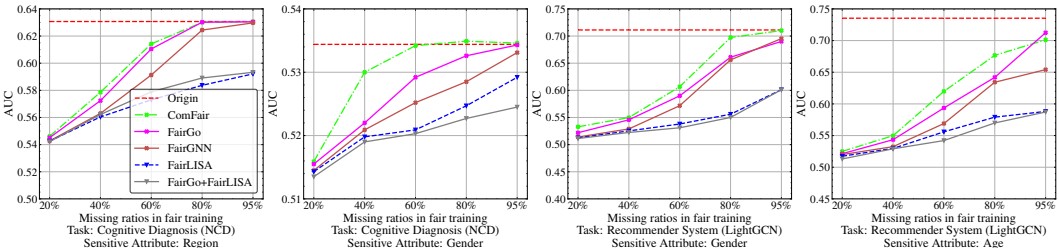

Figure 2: Fairness performance on different missing sensitive attributes ratios ( the lower, the fairer).

ratio reaches 95%, several fairness-aware baselines converge to the Origin baseline. This observation indicates that in the absence of extremely sensitive attributes, other baseline methods have essentially lost their effectiveness in achieving fairness and perform similarly to the case where fairness was not considered at all. This highlights the point that fair user modeling in limited sensitive situations is worth studying.

- The models that can deal with unknown sensitive attributes, i.e., FairGNN, FairLISA, FairGo+FairLISA, perform better in most cases. It shows that making use of data with unknown sensitive attributes can alleviate the limited known sensitive information problem to some extent.

- Our methods, i.e., FairLISA and FairGo+FairLISA, consistently achieve the best performance. As the missing ratio increases, the improvements of our methods over FairGNN become more prominent. When the missing ratio reaches 95%, FairLISA can achieve an 8% improvement in fairness compared to FairGNN. This is due to FairGNN's reliance on the accuracy of the estimator. When numerous sensitive attributes are present, the accuracy of the sensitive estimator can be ensured. Nevertheless, FairGNN encounters notable constraints when access to sensitive information is limited. Our methods, based on information theory, can directly apply data with unknown sensitive attributes to improve fairness and are not greatly affected by the missing ratios, demonstrating the robustness of our methods.

**Influence of $U_L$, $U_N$ (RQ3).** FairLISA can directly leverage $U_L$ and $U_N$ to improve fairness. In this part, we investigate their influence by adjusting the hyperparameter $\lambda_2$ ($U_L$), $\lambda_3$ ($U_N$) as defined in Eq. (9) on two user models (i.e., NCD, LightGCN). Additional results are included in Appendix C.5. Specifically, we vary the values of $\lambda_2$, $\lambda_3$ on NCD as {0, 0.5, 1, 1.5} and the values of $\lambda_2$, $\lambda_3$ on LightGCN as {0, 5, 10, 15}. The results are shown in Figure 5 (the lower, the fairer). We can find that $\lambda_2$ has a greater impact than $\lambda_3$, indicating that $U_L$ has a

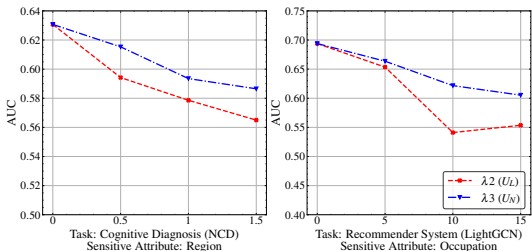

Figure 3: Effects of $\lambda_2$, $\lambda_3$ on user models.

more significant role in promoting fairness. This is reasonable because sensitive labels in $U_L$ can guide the filtering process. Moreover, as the values of hyperparameters increase, the results become fairer, suggesting that both $U_L$ and $U_N$ contribute to fairness.

**Relation to classical group fairness metrics (RQ4).** Previous experiments have demonstrated the effectiveness of FairLISA in terms of preventing the leakage of sensitive information from representations. In this section, we present the results of our proposed model on two classical group fairness metrics, namely Demographic Parity [7] and Equal Opportunity [13] (the lower, the better). The specific formulas for these metrics and additional results are included in Appendix C.6. In short, the results shown in Figure 7 indicate that our goal of removing the effect of sensitive attributes

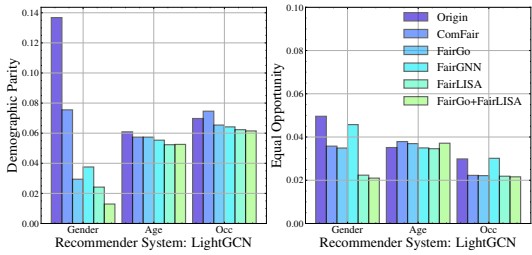

Figure 4: Performance of different group fairness metrics (the lower, the better).

also benefits the classical group fairness metrics, and our models achieve the best performance.

# 6 Conclusion

In this paper, we focused on the fairness problem of user modeling and proposed a general framework FairLISA as a solution in limited sensitive information situations. We first provide a novel theoretical perspective to unify data with known and unknown sensitive attributes with the fair mutual information objective. Building on this unification, we propose an adversarial learning-based framework that enables us to effectively leverage the entire user dataset for fair user modeling. We further showed that FairLISA could be seen as a generalized framework from traditional adversarial methods and take better advantage of data with unknown sensitive attributes. Experimental results on different user modeling tasks showed the effectiveness of FairLISA.

# 7 Border Impact and Future Work

User modeling is a crucial task in various applications and has a significant impact on various aspects of our daily lives. In this paper, we discuss the social responsibilities of user modeling and consider a situation where limited sensitive attributes are available. Our model could better help to design fairness-aware user modeling tasks and also protect users' privacy in the user modeling process. However, we notice that despite considering a limited sensitive situation, there is still a need to collect part of users' private information as model input, which may increase the burden on users as well as incurs possible privacy leakage in the data storage and transfer process. In the future, we will try to combine the privacy and fairness concerns so as to explore fairness and privacy aware user modeling.

## Acknowledgments and Disclosure of Funding

This research was partially supported by grants from the National Key Research and Development Program of China (Grant No. 2021YFF0901003) and the National Natural Science Foundation of China (Grants No. 62337001, U20A20229).

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

**Appendix**

In the appendix, we start from the proofs of Lemma 4.1 and Lemma 4.2 in Section A. After that, we introduce the whole optimization algorithm for FairLISA in Section B. Finally, the supplementary experimental introduction and results are reported in Section C.

## A  Proof

FairLISA is an adversarial learning-based framework designed to promote fair model training by efficiently utilizing data with known and unknown sensitive attributes. It consists of two modules: a filter module and a discriminator module. Specifically, the filter module $\mathcal{F}$ is trained to filter out the information of sensitive attributes (e.g., gender) of $\theta$. We denote the filtered user modeling results of a user $u$ as: $\hat{\theta} = \mathcal{F}(\theta)$. The discriminator module $\mathcal{D}$ tries to predict the corresponding sensitive attribute from the filtered user representations. The fundamental concept behind FairLISA is to establish a connection between two types of data and the fairness objective (i.e., $I(\hat{\theta}, S) = 0$) through Lemma 4.1 and Lemma 4.2. By doing so, it effectively utilizes data with both known and unknown sensitive attributes to facilitate fair model training. In the following sections, we will provide a detailed explanation of Lemma 4.1 and Lemma 4.2, respectively.

### A.1  Lemma 4.1

In this proof, we explain how FairLISA builds the relationship between fairness goal $I(\hat{\theta}, S) = 0$ with user data with known sensitive attributes $U_L$ through Lemma A.1. We define $\mathcal{D}_s(\hat{\theta})$ as the probability that the discriminator $\mathcal{D}$ recognizes the value of the sensitive attribute $S$ as $s$ according to $\hat{\theta}$, $p(\hat{\theta})$ be the distribution of users' filtered representations.

**Lemma A.1.** *Given the optimal discriminator $\mathcal{D}_s^*(\hat{\theta})$ and let $p(\hat{\theta}|s)$ be the distribution of users' filtered representations whose sensitive attribute is $s$, then $I(\hat{\theta}, S)$ can be derived as*

$$I(\hat{\theta}, S) = \sum_s p(s)\mathbb{E}_{\hat{\theta}\sim p(\hat{\theta}|s)}[\log \mathcal{D}_s^*(\hat{\theta})] + H(S). \tag{11}$$

*Proof.* We use $\mathcal{L}_D$ represents the loss of discriminator $D$, we first prove that when $\mathcal{L}_D = -\sum_s p(s)\mathbb{E}_{\hat{\theta}\sim p(\hat{\theta}|s)}[\log \mathcal{D}_s(\hat{\theta})]$, the discriminator can achieve the optimal.

Consider the constraints of the discriminator $\sum_s \mathcal{D}_s(\hat{\theta}) = 1$, we can use Lagrange multipliers:

$$\mathcal{L}_{D,\lambda} = -\sum_s p(s)\mathbb{E}_{\hat{\theta}\sim p(\hat{\theta}|s)}\log[\mathcal{D}_s(\hat{\theta})] + \sum_{\hat{\theta}} \lambda_{\hat{\theta}}(\sum_s \mathcal{D}_s(\hat{\theta}) - 1) \tag{12}$$

Differentiate $\mathcal{L}_{D,\lambda}$ with respect to $\mathcal{D}_s(\hat{\theta})$ and $\lambda_{\hat{\theta}}$ and make the corresponding result 0, we get:

$$-p(s)\frac{p(\hat{\theta}|s)}{\mathcal{D}_s(\hat{\theta})} + \lambda_{\hat{\theta}} = 0, \forall s, \hat{\theta}, \qquad \sum_s \mathcal{D}_s(\hat{\theta}) - 1 = 0, \forall \hat{\theta}, \tag{13}$$

therefore, we can calculate that $\mathcal{D}_s^*(\hat{\theta}) = p(s|\hat{\theta})$. Then we substitute it into the right side of the formula and get,

$$\begin{aligned}
&\sum_s p(s)\mathbb{E}_{\hat{\theta}\sim p(\hat{\theta}|s)}[\log \mathcal{D}_s^*(\hat{\theta})] + H(S) \\
&= \sum_s p(s)\mathbb{E}_{\hat{\theta}\sim p(\hat{\theta}|s)}[\log p(s|\hat{\theta})] - \sum_s p(s)\log p(s) \\
&= \int \sum_s p(s)p(\hat{\theta}|s)\log \frac{p(\hat{\theta}|s)}{p(\hat{\theta})}d\hat{\theta} \\
&= \sum_s p(s)KL(p(\hat{\theta}|s)||p(\hat{\theta})) \\
&= JS_{p(s)}(p(\hat{\theta}|s)) = I(\hat{\theta}, S)
\end{aligned} \tag{14}$$

Here, the second last equality is due to the definition of the Jensen-Shannon divergence, and the last equality is due to its equivalence to the mutual information [24]. This completes the proof. $\qquad\square$

We have established a connection between $U_L$ and $I(\hat{\theta}, S)$ through Lemma A.1. Consequently, fairness can be achieved by minimizing Eq. (11) with a fixed optimal discriminator $\mathcal{D}^*$.

## A.2 Lemma 4.2

In this proof, we explain how FairLISA builds the relationship between fairness goal $I(\hat{\theta}, S) = 0$ with user data with known sensitive attributes $U_N$ through Lemma A.1.

**Lemma A.2.** *Given the optimal discriminator $\mathcal{D}_s^*(\hat{\theta})$ and let $p(\hat{\theta})$ be the distribution of users' filtered representation, we have,*

$$
\begin{aligned}
I(\hat{\theta}, S) &= \mathbb{E}_{\hat{\theta} \sim p(\hat{\theta})} \sum_s \mathcal{D}_s^*(\hat{\theta}) \log \mathcal{D}_s^*(\hat{\theta}) + H(S) \\
&= -H(\mathcal{D}_s^*(\hat{\theta})) + H(S).
\end{aligned}
\tag{15}
$$

*Proof.* According to the relationship between mutual information and JS divergence, the relationship between JS divergence and KL divergence [24], we have

$$
I(\hat{\theta}, S) = JS_{p(s)}(p(\hat{\theta}|s)) = \sum_s p(s) KL(p(\hat{\theta}|s)||p(\hat{\theta})).
\tag{16}
$$

By calculating KL divergence, we get,

$$
\sum_s p(s) KL(p(\hat{\theta}|s)||p(\hat{\theta})) = \int \sum_s p(s) p(\hat{\theta}|s) \log \frac{p(\hat{\theta}|s)}{p(\hat{\theta})} d\hat{\theta}.
\tag{17}
$$

Then we expand the formula and use bayes' theorem,

$$
\begin{aligned}
&\int \sum_s p(s) p(\hat{\theta}|s) \log \frac{p(\hat{\theta}|s)}{p(\hat{\theta})} d\hat{\theta} \\
&= \int \sum_s p(s) p(\hat{\theta}|s) \log \frac{p(s) p(\hat{\theta}|s)}{p(\hat{\theta})} d\hat{\theta} + H(S) \\
&= \int p(\hat{\theta}) \sum_s \frac{p(s) p(\hat{\theta}|s)}{p(\hat{\theta})} \log \frac{p(s) p(\hat{\theta}|s)}{p(\hat{\theta})} d\hat{\theta} + H(S) \\
&= \int p(\hat{\theta}) \sum_s p(s|\hat{\theta}) \log p(s|\hat{\theta}) d\hat{\theta} + H(S).
\end{aligned}
\tag{18}
$$

Noted that $p(s|\hat{\theta})$ is exactly $\mathcal{D}^*$, we replace it in the formula, and get the result,

$$
\begin{aligned}
&\int p(\hat{\theta}) \sum_s p(s|\hat{\theta}) \log p(s|\hat{\theta}) d\hat{\theta} + H(S) \\
&= \int p(\hat{\theta}) \sum_s \mathcal{D}_s^*(\hat{\theta}) \log \mathcal{D}_s^*(\hat{\theta}) d\hat{\theta} + H(S) \\
&= \mathbb{E}_{\hat{\theta} \sim p(\hat{\theta})} \sum_s \mathcal{D}_s^*(\hat{\theta}) \log \mathcal{D}_s^*(\hat{\theta}) + H(S) \\
&= -H(\mathcal{D}^*(\hat{\theta})) + H(S).
\end{aligned}
\tag{19}
$$

This completes the proof. $\qquad\square$

We have successfully established the relationship between $I(\hat{\theta}, S)$ and $U_N$ through Lemma A.2. Consequently, we can minimize our fairness criterion $I(\theta, S)$ directly by minimizing Eq. (15) using $U_N$ without the need to predict unknown sensitive attribute labels.

---
**Algorithm 1:** The FairLISA framework
---
**Input:** Training user set $U$; item set $V$; interaction records $R$; Filter $\mathcal{F}$; Discriminator $\mathcal{D}$;
   Training epochs $M$; Discriminator training steps $T$; hyperparameters $\lambda_1, \lambda_2, \lambda_3$
Initialize: User modeling results $\theta, \forall \theta \in U$
**for** *epoch* $\longleftarrow$ *1 to M* **do**
   **for** $u \in U$ **do**
      $\hat{\theta} \longleftarrow \mathcal{F}(\theta)$;
      Optimize $\mathcal{F}$ through $\mathcal{L}_{\mathcal{F}}$ in Eq.(9) with $\mathcal{D}$ fixed;
      **for** $t \longleftarrow$ *1 to T* **do**
         Optimize $\mathcal{D}$ through $\mathcal{L}_{\mathcal{D}}$ in Eq.(10) with $\theta$, $\mathcal{F}$ fixed;
      **end**
   **end**
**end**
---

## B   Training Algorithm for FairLISA

## C   Supplementary Experimental Results

### C.1   User Modeling Tasks

To verify the effectiveness of our model, we choose two representative issues in real-world user modeling scenarios. The first is user capability modeling in areas such as education which is called cognitive diagnosis [40]. We choose several representative models:

- IRT [30] is a cognitive diagnosis method that models the cognitive processes from students' exercise records with a logistic-like function.
- MIRT [4] is a variant of the basic IRT model, which extends the unidimensional latent traits of students and exercises in IRT to multidimensional vectors.
- NCD [41] is a deep neural cognitive diagnosis framework that models the complex interaction from students' exercising records with a multilayer perceptron (MLP).

The second is user preference modeling in recommender system [39]. We choose several representative models:

- PMF [34] is the Probabilistic Matrix Factorization algorithm by adding Gaussian prior distribution of user and item latent factors into the classical matrix factorization algorithm.
- NCF [15] is a state-of-the-art collaborative filtering model based on deep neural networks.
- LightGCN [16] is a state-of-the-art GCN-based recommender system model which learns user embeddings by linearly propagating them on the user-item interaction graph.

### C.2   Dataset Description

PISA2015[4] is released by Program for International Student Assessment (PISA) , which collects students' information from over one hundred regions or countries. It contains 403,609 students, 183 exercises, and the records between students and exercises are 11,488,234. We select region, gender, family education, and family economic as sensitive attributes. For region, users are divided into two groups according to whether users' region belongs to Organisation for Economic Co-operation and Development (OECD) countries. According to whether the highest education degree of parents is above primary level, we divide users into two groups. Similarly we do so according to whether there are houses belonging to users. More details of the PISA dataset can be found in [52].

MovieLens-1M[5] is a benchmark dataset for recommender systems. It contains 1 million rating records for about 4,000 movies from 6,040 users. We select gender, age, and occupation as user

---
[4]https://www.oecd.org/pisa/data/2015database/
[5]https://grouplens.org/datasets/movielens/1m/

Table 2: The fairness performance of baselines on cognitive diagnosis task (AUC scores of all attackers). R, A, E, C represent region, age, family education, family economic. The smaller values of AUC denote better fairness performance with less sensitive information leakage (the fairer). The best results are highlighted in bold.

| | IRT | | | | MIRT | | | | NCD | | | |
|---|---|---|---|---|---|---|---|---|---|---|---|---|
| | AUC-R | AUC-A | AUC-E | AUC-C | AUC-R | AUC-A | AUC-E | AUC-C | AUC-R | AUC-A | AUC-E | AUC-C |
| Origin | 0.5864 | 0.5191 | 0.5930 | 0.5565 | 0.6271 | 0.5351 | 0.6329 | 0.5831 | 0.6307 | 0.5344 | 0.6327 | 0.5808 |
| ComFair | 0.5661 | 0.5178 | 0.5858 | 0.5515 | 0.5727 | 0.5226 | 0.5806 | 0.5618 | 0.5786 | 0.5300 | 0.5795 | 0.5609 |
| FairGo | 0.5699 | 0.5179 | 0.5828 | 0.5489 | 0.5700 | 0.5201 | 0.5965 | 0.5624 | 0.5724 | 0.5220 | 0.5950 | 0.5597 |
| FairGNN | 0.5468 | 0.5165 | 0.5452 | 0.5284 | 0.5680 | 0.5181 | 0.5952 | 0.5562 | 0.5761 | 0.5209 | 0.5944 | 0.5591 |
| FairLISA | 0.5412 | 0.5163 | 0.5490 | 0.5279 | 0.5540 | 0.5154 | 0.5727 | 0.5429 | **0.5540** | **0.5154** | 0.5727 | **0.5429** |
| FairGo+FairLISA | **0.5355** | **0.5152** | **0.5419** | 0.5279 | **0.5510** | **0.5154** | **0.5720** | **0.5422** | 0.5622 | 0.5190 | **0.5704** | 0.5513 |

Table 3: The accuracy performance of baselines on cognitive diagnosis task. Bold represents the best results, and underline represents runner-up results.

| | IRT | | | | MIRT | | | | NCD | | | |
|---|---|---|---|---|---|---|---|---|---|---|---|---|
| | ACC | AUC | MAE | MSE | ACC | AUC | MAE | MSE | ACC | AUC | MAE | MSE |
| Origin | **0.6932** | **0.7475** | 0.3932 | **0.2042** | **0.7152** | **0.7868** | **0.3529** | **0.1903** | **0.7486** | **0.8111** | **0.3555** | **0.1704** |
| ComFair | 0.6834 | 0.7380 | 0.3953 | 0.2095 | 0.7141 | 0.7864 | 0.3607 | 0.1887 | 0.7253 | 0.8000 | 0.3605 | 0.1816 |
| FairGo | 0.6853 | 0.7422 | 0.3996 | 0.2072 | 0.7147 | 0.7804 | 0.3618 | 0.1897 | 0.7299 | 0.8019 | 0.3597 | 0.1799 |
| FairGNN | 0.6853 | 0.7423 | 0.3972 | 0.2084 | 0.7110 | 0.7857 | 0.3616 | 0.1893 | 0.7301 | 0.8025 | 0.3584 | 0.1800 |
| FairLISA | 0.6923 | 0.7469 | 0.3942 | 0.2044 | 0.7148 | 0.7867 | 0.3565 | 0.1896 | 0.7300 | 0.8039 | 0.3572 | 0.1796 |
| FairGo+FairLISA | 0.6905 | 0.7411 | **0.3927** | 0.2045 | 0.7140 | 0.7864 | 0.3590 | 0.1897 | 0.7313 | 0.8041 | 0.3573 | 0.1793 |

sensitive features, where gender is a binary feature, occupation is a 21-class feature, and users are assigned to 7 groups based on age.

## C.3 Fairness-aware Baselines

we compare FairLISA with the following baselines:

- Origin: basic user models without fairness consideration.

- ComFair [3] is a classical fairness-aware adversarial method in fully sensitive attributes situations.

- FairGo [45] is the state-of-the-art method to improve fairness in fully sensitive attribute situations from a graph perspective.

- FairGNN [5] is the state-of-the-art fairness-aware method in limited sensitive situations relying on a sensitive attribute predictor.

## C.4 Result of Cognitive Diagnosis

The fairness and accuracy results of cognitive diagnosis task are shown in Table 2, Table 3. We have several observations from the out outcomes.

- From the perspective of model fairness (i.e., Table 2), the AUC of attackers on origin user modeling models are significantly higher than 0.5, which means that the sensitive information is exposed by user modeling results even such information is not explicitly used. Further, all fairness-aware methods can improve fairness. Among these methods, models which can simultaneously deal with the known and unknown sensitive attributes information (i.e., FairGNN, FairLISA, FairGo+FairLISA) get better fairness performance. It shows that data with unknown sensitive attributes also benefit fair user modeling. More importantly, we find our methods (i.e., FairLISA, FairGo+FairLISA) outperform others, suggesting the effectiveness of our work.

- From the perspective of model accuracy (i.e., Table 3), all fairness-improving methods lose some accuracy compared with origin user models. This is the inevitable result of the trade-off between accuracy and fairness [3, 33, 45]. The reason is that the fairness-aware methods are aiming at filtering out the information of certain sensitive features from user modeling results, which will to some extent reduce the information contained, thus decreasing accuracy. Among these fairness-aware methods, we could find FairLISA, FairGo+FairLISA have the best accuracy performance, and they could even maintain comparable accuracy with origin models in some cases.

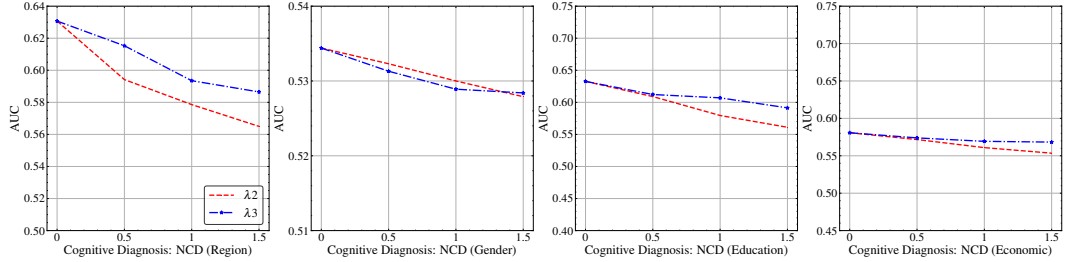

Figure 5: Effects of different hyperparameters on NCD.

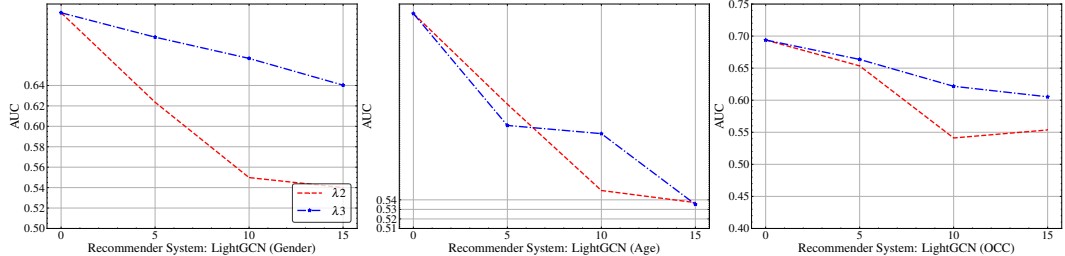

Figure 6: Effects of different hyperparameters on LightGCN.

- From the perspective of balancing the model accuracy and fairness. Our methods can achieve both better accuracy performance and fairness promotion in all cases. Hences, we argue that our framework achieves better performance in balancing the model accuracy and fairness, which also suggests that making full use of data with unknown sensitive attributes can alleviate the problem of fairness-accuracy trade-off to some extent.

## C.5 Influence of $U_L$, $U_N$ (RQ3)

FairLISA unifies the treatment of data with known sensitive attributes $U_L$ and data with unknown sensitive attributes $U_N$ from a mutual information perspective, thus $U_L$, $U_N$ can be directly leveraged to improve fairness. In this part, we investigate their influence on fair user modeling results by adjusting the hyperparameter $\lambda_2$ ($U_L$), $\lambda_3$ ($U_N$). As for the user models, we choose the representative models which achieve the best performance results in RQ1 (i.e., NCD, LightGCN). More specifically, we vary the values of $\lambda_2$, $\lambda_3$ on NCD as {0, 0.5, 1, 1.5} and the values of $\lambda_2$, $\lambda_3$ on LightGCN as {0, 5, 10, 15}. The results are shown in Figure 5 (the lower, the fairer). On the one hand, we find $\lambda_2$ always has a huge impact than $\lambda_3$, which means $U_L$ makes a greater contribution to fairness than $U_N$. This is reasonable because of the sensitive attribute labels in $U_L$ being as the guidance to filter the effect of sensitive attributes. On the other hand, as the values get larger, both of the results get fairer, which indicates that both $U_L$ and $U_N$ can contribute to fairness. This observation also demonstrates that making use of $U_N$ can improve the fairness of the user modeling result.

## C.6 Relation to classical group fairness metrics (RQ4)
Previous experiments have shown the effectiveness of our framework in terms of avoiding leaking sensitive information from user modeling results. In this part, we show the results of our proposed

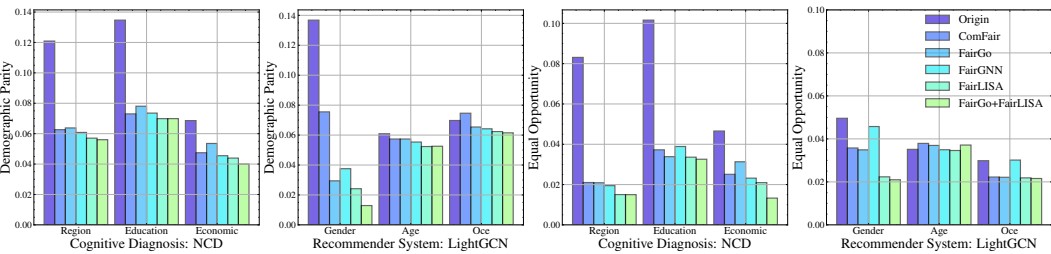

Figure 7: Performance of group fairness metrics on NCD and LightGCN (the lower, the better).

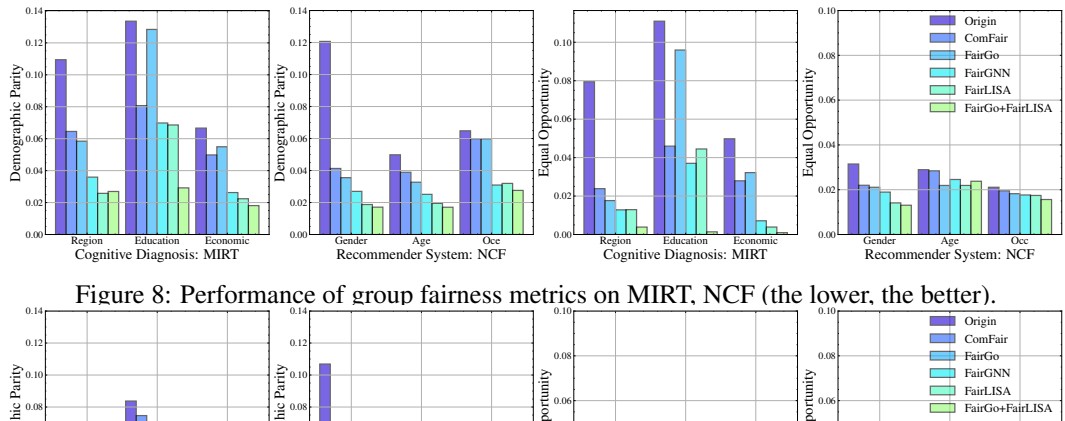

Figure 8: Performance of group fairness metrics on MIRT, NCF (the lower, the better).

Figure 9: Performance of group fairness metrics on IRT, PMF (the lower, the better).

model on classical fairness metrics. We choose two group fairness metrics (i.e., Demographic Parity [7], Equal Opportunity [13]) to evaluate the results. Demographic Parity [7] is widely used to measure the predicted rating discrepancy for binary valued sensitive attributes [45]. Take binary attribute (i.e., gender) for example, the detailed formula is as $DP = \frac{1}{N} \sum_{v=1}^{N} ||E_{u \in male}[\hat{r}_{uv}] - E_{u \in female}[\hat{r}_{uv}]||$. For attributes with multiple values, we borrow the idea of demographic parity and divide users into different groups based on their sensitive values. Then we take the standard deviation of predicted ratings of each user group to measure demographic parity. Equal Opportunity [13] is another classical fairness definition, it advances demographic parity fairness by considering the parity of prediction accuracy of each group. For binary attribute, take gender for example, the detailed formulas are as $EO = \frac{1}{N} \sum_{v=1}^{N} ||E_{u \in male}[\hat{r}_{uv} - r_{uv}] - E_{u \in female}[\hat{r}_{uv} - r_{uv}]||$. For attributes with multiple values, it has the same way as demographic parity. The results of these two group fairness metrics on the user models are shown in Figure 7, Figure 8, Figure 9. We can find both user modeling tasks are unfair on these two classical metrics, which also indicates the necessity to explore fairness in user modeling. Then, all fairness-aware methods aiming to remove the effect of sensitive attributes can improve the fairness of the two metrics. It shows that the goal of removing the effect of sensitive attributes also benefits the classical group fairness metrics. Among all these methods, we could find that the methods (FairGNN, FairGo+FairLISA, FairLISA) can always achieve better results, which shows that it is necessary to explore fairness in limited situations. Meanwhile, we find FairLISA can achieve the best performances, suggesting the effectiveness of our methods.

