# OpenReview forum: "FairLISA: Fair User Modeling with Limited Sensitive Attributes Information"
_NeurIPS.cc/2023/Conference — NeurIPS 2023 poster_

### Official Review · Reviewer_QZCP · 2023-07-05

**Soundness:** 3 good
**Presentation:** 4 excellent
**Contribution:** 3 good
**Rating:** 7
**Confidence:** 4

**Summary:**

This paper aims to achieve fair user modeling with limited sensitive attribute information and propose a general framework, FairLISA, which efficiently utilizes data with known and unknown sensitive attributes to facilitate fair model training. The authors also provide theoretical guarantees from a mutual information perspective. Extensive experiments are conducted to demonstrate the effectiveness of FairLISA in scenarios with different ratios of missing sensitive attributes.


**Strengths:**

S1. The scenario of missing sensitive attributes in fair user modeling is both meaningful and worthy of investigation.

S2. The paper is well-written and easy to follow.

S3. The proposed FairLISA efficiently leverages unknown data without the need for predicting missing attributes, providing a simple yet effective approach, supported by theoretical guarantees.

S4. Extensive experiments, especially the RQ2 experiment on different missing ratio situations, demonstrate the effectiveness and robustness of FairLISA in two representative user modeling tasks.

**Weaknesses:**

W1. This work mainly focuses on the fairness definition where the mutual information between the user modeling result and the sensitive information is zero. However, there are other classic fairness definitions, such as Equalized Odds (EO) and Demographic Parity (DP). It would be valuable to discuss investigate how FairLISA performs on these metrics.

W2. The sensitive information ratio setting is missing in Table 1.

**Questions:**

Q1. How does FairLISA perform on classic fairness metrics?

Q2. Please provide the complete experiment setting, including the missing ratio in RQ1.

**Limitations:**

Although this paper addresses the issue of limited sensitive attribute information, it still requires the collection of some sensitive information as model input, which can potentially compromise privacy. Exploring the combination of fairness and privacy would be a promising direction for future research.

---

> ### Author Rebuttal · Authors · 2023-08-09
>
> >**A1. How does FairLISA perform on classic fairness metrics?**
>
>  Q2: We have already conducted this experiment in our paper, and the details can be inferred from Experiments RQ4 and Appendix C.6. To provide a clearer presentation of these results, we report the performance of all methods on classical fairness metrics for PMF, LightGCN (representative user models in Recommender System) in Table 1, Table 2 (the lower, the better). The results indicate that our goal of removing the effect of sensitive attributes also benefits the classical group fairness metrics, and our models achieve the best performance. We will highlight this experiment in the revision.
>
> Table 1: The performance on classic fairness metrics for PMF (the lower, the better), the best fairness results are highlighted in bold.
>
> |                 | GENDER DP | GENDER EO | AGE DP   | AGE EO   | OCC DP   | OCC EO   |
> | --------------- | --------- | --------- | -------- | -------- | -------- | -------- |
> | Origin          | 0.106871  | 0.031829  | 0.051489 | 0.027891 | 0.060231 | 0.024581 |
> | ComFair       | 0.054697  | 0.026501  | 0.037224 | 0.028744 | 0.047889 | 0.019412 |
> | FairGo         | 0.029125  | 0.020568  | 0.037694 | 0.028481 | 0.041344 | 0.019123 |
> | FairGNN         | 0.027538  | 0.022540  | 0.023698 | 0.027001 | 0.040135 | 0.019478 |
> | FairLISA | 0.014163  | 0.019857  | 0.020140 | 0.024576 | 0.039625 | **0.018324** |
> | FairGo+FairLISA     | **0.010912**  | **0.017749**  | **0.019480** | **0.023984** | **0.039013** | 0.018352 |
>
> Table 2: The performance on  classic fairness metrics for LightGCN (the lower, the better), the best fairness results are highlighted in bold.
> |                 | GENDER DP | GENDER EO | AGE DP   | AGE EO   | OCC DP   | OCC EO   |
> | --------------- | --------- | --------- | -------- | -------- | -------- | -------- |
> | Origin          | 0.136796  | 0.049581  | 0.060878 | 0.035130 | 0.069747 | 0.029848 |
> | ComFair       | 0.075487  | 0.035767  | 0.057361 | 0.037898 | 0.074596 | 0.022252 |
> | FairGo         | 0.029414  | 0.034892  | 0.057358 | 0.036956 | 0.065387 | 0.022143 |
> | FairGNN         | 0.037538  | 0.045751  | 0.055342 | 0.034956 | 0.064192 | 0.030140 |
> | FairLISA | 0.024163  | 0.022345  | 0.052356 | **0.034596** | 0.062334 | 0.021860 |
> | FairGo+FairLISA      | **0.012912**  | **0.020985**  | **0.052539** | 0.037129 | **0.061489** | **0.021578** |
>
>
>
>
> >**A2. Please provide the complete experiment setting, including the missing ratio in RQ1.**
>
> Q2: Sorry for missing the detail. We will include this information in the revision.

---

> > ### Comment · Reviewer_QZCP · 2023-08-19
> > **Thanks for your response**
> >
> > Thanks for the authors’ responses. Most of my concerns have been addressed.
> > Specifically, the experiment was conducted to validate the effectiveness of FairLISA in classic fairness metrics. This should be added in the revision. Furthermore, the authors claims that “we will try to combine the privacy and fairness concerns so as to explore fairness and privacy aware user modeling. ” Though this can be future work, I would to hear from the authors about the plan in some details.

---

> > > ### Author Response · Authors · 2023-08-20
> > >
> > > Dear Reviewer QZCP,
> > >
> > > We sincerely appreciate your recognition and the invaluable feedback you have provided. We  will present our plan for integrating privacy and fairness considerations within FairLISA from pre-processing, in-processing, and post-processing stages.
> > >
> > > In the pre-processing stage, since FairLISA needs collect some sensitive attributes from certain users. To align with stringent privacy requirements, we intend to employ various techniques during the data collection phase. These techniques may encompass actions such as the removal or obfuscation of personally identifiable information (PII) from the dataset, thereby safeguarding individual privacy.
> > >
> > > In the in-processing stage, we will combine some differential privacy mechanisms in FairLISA  to fulfill the requirements of both fairness and privacy. Sepcifically, we will introduce controlled noise to the training data or gradients, guaranteeing that individual data points remain confidential.
> > >
> > > In the post-processing stage, we will assess the model's fairness and privacy performance using appropriate metrics (such as Differential Privacy [1], demographic parity [2], equal opportunity [3]). If any discrepancies or concerns are identified, re-evaluate and fine-tune the model accordingly.
> > >
> > > Once again, we express our gratitude for your valuable feedback. Please feel free to contact me if any other confusion exists.
> > >
> > > [1] Dwork C, McSherry F, Nissim K, et al. Calibrating noise to sensitivity in private data analysis[C]//Theory of Cryptography: Third Theory of Cryptography Conference, TCC 2006, New York, NY, USA, March 4-7, 2006. Proceedings 3. Springer Berlin Heidelberg, 2006: 265-284.
> > >
> > > [2] Cynthia Dwork, Moritz Hardt, Toniann Pitassi, Omer Reingold, and Richard Zemel. Fairness
> > > through awareness. In Proc. of ITCS, 2012.
> > >
> > > [3] Moritz Hardt, Eric Price, and Nati Srebro. Equality of opportunity in supervised learning. Advances in neural information processing systems, 29, 2016.
> > >
> > > Best,
> > >
> > > Authors.

---

### Official Review · Reviewer_xseN · 2023-07-05

**Soundness:** 3 good
**Presentation:** 3 good
**Contribution:** 3 good
**Rating:** 8
**Confidence:** 4

**Summary:**

The authors investigate the problem of fair user modeling in a setting with limited sensitive attributes. Due to the lack of such attribute information, they propose a general framework called FairLISA, which efficiently applies unlabeled data to facilitate fair model training. Compared to previous works, FairLISA can directly leverage unlabeled data without the need for predicting missing attributes, thereby reducing information loss caused by predictions. The experiments demonstrate the effectiveness of the proposed model.

**Strengths:**

The authors tackle a socially valuable problem of fair user modeling in a setting with limited sensitive attributes, which holds significant implications across various applications, such as recommendation systems and cognitive diagnostics.

The paper provides a thorough and insightful summary of related works on fairness without sensitive attributes and fairness in limited sensitive attribute situations. Building upon existing research, this paper effectively identifies and summarizes three key challenges: "Efficient data utilization," "Theoretical guarantee," and "Framework generalization." The authors propose reasonable solutions to address these challenges. As far as I am concerned, the insights of directly leveraging unlabeled data without predicting missing attributes are straightforward and effective.

The authors substantiate their claims with theoretical guarantees through Lemma 1 and Lemma 2. Finally, comprehensive experiments are conducted on two representative tasks, highlighting the superiority of FairLISA. In summary, this work exhibits reasonable motivation, a clear literature review, a look-nice model design, and a comprehensive validation experiments.

**Weaknesses:**

There have been existing works that focus on the complete absence of sensitive attributes, such as [1][2]. Although the specific problems and input data may differ, it would be valuable to explore whether FairLISA can be combined with these works to extend their potential in limited scenarios. This article lacks discussion in these two aspects.

[1]Tianxiang Zhao, Enyan Dai, Kai Shu, and Suhang Wang. Towards fair classifiers without sensitive attributes: Exploring biases in related features. In Proceedings of the Fifteenth ACM International Conference on Web Search and Data Mining, 2022

[2] Tatsunori Hashimoto, Megha Srivastava, Hongseok Namkoong, and Percy Liang. Fairness without demographics in repeated loss minimization. In International Conference on Machine Learning, 2018.

**Questions:**

Can fairness be extended to settings without sensitive attributes?

Can the techniques used in settings without sensitive attributes [1][2] be integrated with FairLISA to attain improved outcomes in limited situations?

[1]Tianxiang Zhao, Enyan Dai, Kai Shu, and Suhang Wang. Towards fair classifiers without sensitive attributes: Exploring biases in related features. In Proceedings of the Fifteenth ACM International Conference on Web Search and Data Mining, 2022

[2] Tatsunori Hashimoto, Megha Srivastava, Hongseok Namkoong, and Percy Liang. Fairness without demographics in repeated loss minimization. In International Conference on Machine Learning, 2018.

**Limitations:**

As mentioned in the Border Impact section, FairLISA is effective only in limited scenarios. It is recommended to further explore the potential of FairLISA in completely unsupervised settings.

---

> ### Author Rebuttal · Authors · 2023-08-09
>
> >**A1. Can fairness be extended to settings without sensitive attributes?**
>
> Q1: I sincerely appreciate your question. Currently, our methods cannot be extended to settings without sensitive attributes due to the necessity of labeled data to train the discriminator. However, it is essential to note that our research focuses on fairness with limited sensitive attributes. Our main contribution lies in efficiently utilizing data with both known and unknown sensitive attributes to facilitate fair model training. Additionally, we provide theoretical guarantees, and our experiments demonstrate that our approach achieves SOTA performance. Moving forward, we will explore how to extend our work to settings without sensitive attributes.
>
>
> >**A2. Can the techniques used in settings without sensitive attributes [1][2] be integrated with FairLISA to attain improved outcomes in limited situations?**
>
> Q2: Thank you for your question. The techniques used in settings without sensitive attributes can indeed be integrated with FairLISA, as exemplified by the related work FairRF[1]. FairRF does not explicitly require sensitive attributes and enhances fairness through related features. Building on this core idea, we can seamlessly combine FairRF and FairLISA. Specifically, by leveraging related features, we can infer high-quality labels for sensitive attributes. This data enables us to train our discriminator and filter. Subsequently, these two components can be optimized jointly, potentially leading to improved results. In the future, we will explore how to integrate our method with existing techniques used in settings without sensitive attributes, aiming to achieve improved fairness and performance results. Meanwhile, we will provide more specific discussions in the revision to enhance the clarity and depth of our work.
>
> [1]Tianxiang Zhao, Enyan Dai, Kai Shu, and Suhang Wang. Towards fair classifiers without sensitive attributes: Exploring biases in related features. In Proceedings of the Fifteenth ACM International Conference on Web Search and Data Mining, 2022
>
> [2] Tatsunori Hashimoto, Megha Srivastava, Hongseok Namkoong, and Percy Liang. Fairness without demographics in repeated loss minimization. In International Conference on Machine Learning, 2018.

---

> > ### Comment · Reviewer_xseN · 2023-08-19
> > **Thanks for your response**
> >
> > Thanks for your response. My concerns have been well addressed. Moreover, this paper holds the potential to make an impact in the situation of fairness-aware with limited sensitive attributes situations. I would raise my score accordingly.

---

### Official Review · Reviewer_Gbnv · 2023-07-08

**Soundness:** 3 good
**Presentation:** 3 good
**Contribution:** 3 good
**Rating:** 5
**Confidence:** 4

**Summary:**

This paper propose FairLISA that learns fair user modeling using limited sensitive attribute information. Specifically, for users with known sensitive attribute information, FairLISA maximizes the cross-entropy of predicting the sensitive attribute using user representations; for users with unknown sensitive attribute, FairLISA maximizes the entropy of predicting sensitive attribute using user representations. Experiments on benchmark datasets demonstrate the efficacy of the proposed model.


**Strengths:**

S1. Studying limited sensitive attribute scenario is practical and important.

S2. FairLISA is supported by the theoretical analysis.

S3. FairLISA's performance is good based on the empirical evaluation.


**Weaknesses:**

Please see limitations.


**Questions:**

Please see limitations.


**Limitations:**

L1. The authors may better illustrate why statistical parity is important in recommender system. To my understanding, many recommendation tasks is related to sensitive attribute as well. For example, a recommender system don't want to recommend feminine care items to male users.

L2. Using user modeling as a motivating example is ok to me, but I don't understand why the authors position the paper to fair user modeling specifically. How does the proposed method connect to user modeling? What is the uniqueness of user modeling (other than limited sensitive attribute) in terms of fair user embedding learning? How does <user, item, relation> triplet useful here?

L3. The theoretical analysis is based on the fact that discriminator is Bayesian optimal, which is often impossible. Thus, the practicability of the theoretical analysis needs more justification.

L4. What if the discriminator is not optimal? Does the theoretical analysis still hold in this case?

L5. Some intuition about the theoretical analysis would strengthen the intuition. For example, how does optimizing the entropy of the discriminator's output help with removing sensitive information. My guess is that it tries to make the prediction probability of sensitive attribute to be uniform. But the authors may clarify it better.

L6. In Figure 2, why do all methods other than FairLISA converge to the same point? And it seems FairLISA converge to the same point when the ratio increases to 95% as well. Some discussion is needed.

L7. FairGNN is developed in the setting of binary sensitive attribute and binary classification, where the covariance regularizer minimize the pseudo-sensitive attribute and the output logit scores. How do the authors extend it to non-binary sensitive attribute settings?

---

> ### Author Rebuttal · Authors · 2023-08-09
>
> >**Q1. The authors may better illustrate why statistical parity is important in recommender system. To my understanding, many recommendation tasks is related to sensitive attribute as well. For example, a recommender system don't want to recommend feminine care items to male users. (L1)**
>
> A1. Sorry for the confusion. Recommender systems have a wide range of applications. In specific recommendation contexts, such as suggesting feminine care items, considering statistical parity may not be relevant since the preferences are inherently linked to gender. However, in other critical scenarios like career recommendations, it becomes essential to incorporate statistical parity. By doing so, we can prevent discriminatory outcomes and ensure fairness for all users, regardless of their sensitive attributes, this goal has been widely adopted the previous works, such as [1][2]. We will reorganize our paper to address your concerns.
>
> >**Q2.  Using user modeling as a motivating example is ok to me, but I don't understand why the authors position the paper to fair user modeling specifically. How does the proposed method connect to user modeling? What is the uniqueness of user modeling (other than limited sensitive attribute) in terms of fair user embedding learning? How does <user, item, relation> triplet useful here? (L2)**
>
> A2. The core of FairLISA is based on the widely adopted framework of fair adversarial learning. In other words, FairLISA can be applied to any domain where fairness adversarial learning is applicable. It is not restricted to specific data formats, such as <user, item, relation>. In this paper, we concentrate on the user modeling domain, as it plays a crucial role in decision-making. Moving forward, we intend to extend the application of FairLISA to a broader range of data formats, exploring its potential in various domains.
>
> >**Q3. Theoretical analysis. (L3,L4)**
>
> A3. The theoretical analysis demands an optimal discriminator assumption. However, I should mention that an optimal discriminator is a commonly ideal assumption in the limited sensitive research domain, such as [4]. Meanwhile, even when the sensitive attributes are not missing, achieving an optimal discriminator is also an ideal scenario. To evaluate the impact of the optimal discriminator assumption on our model, we conducted empirical experiments (RQ2). Specifically, we varied the ratio of missing sensitive attributes in the training set as {20%, 40%, 60%, 80%, 95%}. As the missing ratios increased, it became more challenging for the discriminator to reach an optimal state. By comparing our model's performance with baseline results, we could assess the impact. The experimental results demonstrated that our model achieved SOTA performance, even under the same missing attribute ratios. This indicates that the optimal discriminator assumption has minimal influence on our model compared to the baselines. Thanks for your suggestion and question, we will add more relevant discussions in the revision.
>
> >**Q4. Strengthen the intuition of FairLISA. (L5)**
>
> A4. Thanks for your suggestion. We will illustrate the intuition from the adversary learning perspective.
> Basic fair adversary learning comprises two modules:
> A discriminator module that predicts the sensitive features from the learned user embedding.
> A filter module that aims to fail the discriminator's performance.
> By maximizing the entropy of the discriminator, we can ensure that the prediction probability of sensitive attributes becomes uniform. For example, in the case of gender prediction, both male and female probabilities are set to 0.5. This shows that the discriminator becomes unable to predict the sensitive features from the learned user embedding. Consequently, our goal of achieving fairness is accomplished. We will provide further clarification in the revised version.
>
>
> >**Q5. More discussion about experiments. (L6)**
>
> A5. Thank you for your suggestion. We observe that all methods, except FairLISA, converge to the **Origin** baseline in the missing ratio 95%. **Origin** refers to the model without fairness considerations. This observation indicates that in the absence of extremely sensitive attributes,  other baseline methods have essentially lost their effectiveness in achieving fairness and perform similarly to the case where fairness was not considered at all. In contrast, FairLISA demonstrates a more robust performance, showcasing its effectiveness in achieving fairness. We will provide a more comprehensive discussion on this in the revision.
>
> >**Q6. The non-binary setting expansion of FairGNN. (L7)**
>
> A6. Sorry for missing the detail information.
> The covariance regularizer in FairGNN is
> $$L_R=cov(\hat{s},\hat{y})$$
> while $\hat{s}$ represents the predicted sensitive attribute and $\hat{y}$ represents predicted label.
>
> In the non-binary setting, $\hat{s}$
> s a multi-dimensional vector. To implement the regularization, we first calculate the covariance matrix and then add the frobenius norm of the covariance matrix to the loss function as the regularization term. This technique has been widely adopted in various studies, such as [3]. We will provide a more comprehensive explanation of this setting in the revised version.
>
> [1] Le Wu, Lei Chen, et al. Learning fair
> representations for recommendation: A graph-based perspective. In Proceedings of the Web Conference 2021, 2021
>
> [2] Yunqi Li, Hanxiong Chen, et al.Towards personalized fairness based on causal notion. In Proceedings of the 44th International ACM SIGIR Conference on Research and Development in Information Retrieval, 2021.
>
> [3] Pourahmadi M. Covariance estimation: The GLM and regularization perspectives[J]. 2011.
>
> [4] Enyan Dai and Suhang Wang. Say no to the discrimination: Learning fair graph neural networks with limited sensitive attribute information. In Proceedings of the 14th ACM International Conference on Web Search and Data Mining, 2021.

---

> > ### Comment · Reviewer_Gbnv · 2023-08-17
> > **Response to author rebuttal**
> >
> > Dear authors,
> >
> > I appreciate your efforts in addressing my concerns. But I think my first two concerns remain.
> >
> > (1) Could you provide a more concrete and detailed **real-world** example, in which statistical parity is indeed important and needs to be satisfied?
> >
> > (2) In your response, you mentioned that FairLISA can be applied to other problems. Then why are we specifically learning fair user embedding but overlooking the problem of learning fair more neutral item embedding?
> >
> > Thanks for your efforts in advance.
> >
> > Best,
> >
> > Reviewer Gbnv

---

> > > ### Author Response · Authors · 2023-08-17
> > >
> > > Dear Reviewer Gbnv,
> > >
> > > Thanks for your valuable feedback, we will try our best to alleviate all your concerns, detailed responses are as follows.
> > >
> > > > (1) Could you provide a more concrete and detailed real-world example, in which statistical parity is indeed important and needs to be satisfied?
> > >
> > > Sorry for the confusion. Let's take the career recommendations as an example. Due to historical factors, specific demographic groups (e.g., ethnic minorities and women) may have encountered unjust treatment during the company's recruitment process, placing them at a distinct disadvantage when seeking job opportunities. For example, stereotypes like "Man is to Computer Programmer and Woman is to Homemaker"[1] have perpetuated this bias. To redress these inequalities rooted in historical biases and foster a more inclusive and equitable career development environment, it is crucial to consider the fairness definition of statistical parity.
> > >
> > > In the real world, LinkedIn researchers also argue that recommendation tasks should adhere to statistical parity definition. The top results should always reflect the gender distribution of all candidates [2]. In light of this, they have introduced a fairness-aware algorithm into LinkedIn Talent Search that incorporates statistical parity. This algorithm's effectiveness also has been substantiated through online A/B testing.
> > >
> > > In the revision, we will reorganize our paper to address your concerns.
> > >
> > > > (2) In your response, you mentioned that FairLISA can be applied to other problems. Then why are we specifically learning fair user embedding but overlooking the problem of learning fair more neutral item embedding?
> > >
> > > Thanks for your insightful question. I quite agree that fair item embedding learning is essential in the <user, item, relation> triplet data format, which is also the uniqueness of <user, item, relation>. In fact, the studies of fair recommendation can be classified into two categories based on whether the uniqueness of the <user, item, relation> triplet is utilized: methods do not consider the uniqueness (e.g., ComFair [3], which primarily focuses on fair user embedding), methods consider the uniqueness (e.g., FairGo [4], which studies fairness from the perspective of the user-item interaction graph).
> > >
> > > Our study, however, studies fairness from the limited sensitive attribute perspective.  This stands in parallel with the consideration of the uniqueness in <user, item, relation>.  This implies that regardless of whether the model takes uniqueness into account, we have the ability to extend its application to limited situations. More specifically, our FairLISA can be applied to other adversarial-based models, including ComFair and FariGo.
> > >
> > > In order to validate the efficacy of our model, we have conducted comprehensive investigations across various models and illustrated how we extended these models to limited situations in the paper(as introduced in section 4.5).  Specifically, if we set $\lambda_3$ in Eq. (9) to 0, FairLISA degenerates to ComFair.  If we combine Eq. (8) from FairLISA with the final loss of FairGo. FairGo will be expanded to the limited situation, which we refer to FairGo+FairLISA. Finally, our model's effectiveness was demonstrated across different models. Moreover, our experiments also revealed that FairGo+FairLISA consistently achieves superior fairness outcomes. This implied that by considering the specificity of the <user, item, relation> triplet, we can achieve fairer results in the limited situation.
> > >
> > > Sorry for the confusion. We would reorganize our paper to make it clearer in the revised version. Please feel free to contact me if any other confusion exists.
> > >
> > >
> > > [1] Bolukbasi T, Chang K W, Zou J Y, et al. Man is to computer programmer as woman is to homemaker? debiasing word embeddings[J]. Advances in neural information processing systems, 2016, 29.
> > >
> > > [2] Geyik S C, Ambler S, Kenthapadi K. Fairness-aware ranking in search & recommendation systems with application to linkedin talent search[C]//Proceedings of the 25th acm sigkdd international conference on knowledge discovery & data mining. 2019: 2221-2231.
> > >
> > > [3] Avishek Bose and William Hamilton. Compositional fairness constraints for graph embeddings.  In International Conference on Machine Learning, 2019.
> > >
> > > [4] Le Wu, Lei Chen, et al. Learning fair representations for recommendation: A graph-based perspective. In Proceedings of the Web Conference 2021, 2021
> > >
> > > Best,
> > > Authors.

---

> > > > ### Comment · Reviewer_Gbnv · 2023-08-19
> > > > **Thank you for the detailed response**
> > > >
> > > > I appreciate the authors' efforts in addressing my concern. I agree that FairLISA offers a new way to solve fairness in limited sensitive attribute scenario. But the paper is quite specific in fair *user modeling* with limited sensitive attribute. In that sense, simply formulating the problem as fair representation learning with limited sensitive attribute information is kind of a focus shift in my opinion. Other than that, I do not have further concern and will keep my current evaluation.

---

> > > > > ### Author Response · Authors · 2023-08-20
> > > > >
> > > > > Dear Reviewer Gbnv,
> > > > >
> > > > > Thank you for your valuable suggestions. We will show the potential of FairLISA in other data formats, such as tabular data.
> > > > >
> > > > > Specifically, we have selected a classic adversarial learning method on tabular data, namely AD [1]. Then we have extended its application to scenarios with missing sensitive attributes by leveraging the core principles of FairLISA. More concerely, for data with known sensitive attribute information, FairLISA maximizes the cross-entropy of the discriminator; for users with unknown sensitive attributes, FairLISA maximizes the entropy of the discriminator. Finally, we conduct experiments on the dataset COMPAS, where the goal is to predict recidivism based on the offenders’ features. Here we only consider race as the sensitive attribute and only use samples of African Americans and whites. The evaluation and baselines detail are shown as follows.
> > > > >
> > > > > **Evaluation**
> > > > >
> > > > > For accuracy evaluation, we employ AUC. For fairness evaluation, we employ Statistical Parity [2] (SP) metric.
> > > > >
> > > > > **Baselines**
> > > > >
> > > > > * Origin: a basical MLP model;
> > > > > * AD: the classical fairness-aware adversarial Learning based on MLP model;
> > > > > * FairGNN: a fairness-aware adversarial learning framework in sensitive limited situation, whose core idea is predicting missing sensitive labels based on data with known sensitive attributes. Here, we expand the idea to AD so that it can handle data with missing sensitive attributes.
> > > > >
> > > > >
> > > > > The experimental results are presented in Table 1. It can be observed that FairLISA achieves the best fairness performance compared to the baseline, even in scenarios where 80% of sensitive attributes are missing, while maintaining comparable accuracy outcomes.
> > > > > This also indicates the potential of FairLISA to attain enhanced fairness on tabular data.
> > > > >
> > > > > Table 1: Accuracy and fairness performance overall methods on Compas dataset (The sensitive missing ratio is 80%). AUC represents the accuracy performance(the higher, the better). SP represents the fairness performance(the lower, the better). The best fairness results methods are highlighted in bold.
> > > > > |   Baseline   | AUC | SP |
> > > > > | --------------- | ---------- | ------- |
> > > > > | Origin | 0.7301  | 0.2284 |
> > > > > |AD | 0.6816 | 0.1893 |
> > > > > | FairGNN | 0.6761 | 0.1656 |
> > > > > | FairLISA | 0.6774 | **0.1535** |
> > > > >
> > > > >
> > > > > Thanks again for your valuable suggestion. Please feel free to contact me if any other confusion exists.
> > > > >
> > > > > [1]Zhang B H, Lemoine B, Mitchell M. Mitigating unwanted biases with adversarial learning[C]//Proceedings of the 2018 AAAI/ACM Conference on AI, Ethics, and Society. 2018: 335-340.
> > > > >
> > > > > [2] Cynthia Dwork, Moritz Hardt, Toniann Pitassi, Omer Reingold, and Richard Zemel. Fairness
> > > > > through awareness. In Proc. of ITCS, 2012.
> > > > >
> > > > > Best,
> > > > >
> > > > > Authors.

---

### Official Review · Reviewer_teqm · 2023-07-11

**Soundness:** 3 good
**Presentation:** 3 good
**Contribution:** 3 good
**Rating:** 6
**Confidence:** 3

**Summary:**

This paper proposes a novel adversarial learning method for fairness with limited demographics.

**Strengths:**

Pros:
1. This paper focus on an important and practical problem. Fairness with limited demographics is a practical and important problem.
2. This paper provides a theoretical-driven perspective on fairness with limited demographics.
3. This paper proposed a novel adversarial learning method for fairness with limited demographics.
4. Solid experiments are conducted to demonstrate the effectiveness of the proposed method.


**Weaknesses:**

Cons:
1. The datasets used in this paper seem not very common. So it is a little bit hard to evaluate the effectiveness of the proposed method. It is suggested that authors also conduct experiments on commonly used fairness datasets such as ADULT and COMPAS.
2. Baselines seem insufficient. It is suggested that the authors may consider adding more in-processing methods as baselines.
3. This paper only empirically study the effectiveness of the proposed method. It is unknown whether or not the proposed method is theoretically better than the previous estimation-based method [1]. It is suggested the authors could add some theoretical analysis to demonstrate that the proposed method is better than the previous estimation method.
4. The authors are encouraged to provide code for reproduction.
5. Besides [1], the authors may consider citing some related references on fairness with limited exact demographics. Reference [2] studies the problem of fairness with limited clean sensitive attributes and mostly private sensitive attributes. Reference [3] studies the problem of fairness with active sensitive attribute annotation.

[1] Say no to the discrimination: Learning fair graph neural networks with limited sensitive attribute information. https://arxiv.org/abs/2009.01454

[2] When Fairness Meets Privacy: Fair Classification with Semi-Private Sensitive Attributes https://arxiv.org/abs/2207.08336

[3] Mitigating Algorithmic Bias with Limited Annotations https://arxiv.org/abs/2207.10018


**Questions:**

See Weaknesses.

**Limitations:**

This paper seems to not discuss the limitations.

---

> ### Author Rebuttal · Authors · 2023-08-09
>
> >**Q1. The datasets used in this paper seem not very common. So it is a little bit hard to evaluate the effectiveness of the proposed method. It is suggested that authors also conduct experiments on commonly used fairness datasets such as ADULT and COMPAS.**
>
> A1:We greatly appreciate your suggestion. In this paper, we study the fairness in **user modeling**, where the fundamental data format is (user, item, relation). This format is distinct from the ADULT and COMPAS table data formats, rendering these widely used datasets unsuitable for our purposes. Instead, we have chosen to utilize the classical fairness datasets commonly employed in user modeling studies [1][2], such as MovieLens-1M.
>
> >**Q2. Baselines seem insufficient. It is suggested that the authors may consider adding more in-processing methods as baselines.**
>
> A2: Thanks for your valuable suggestion. We have incorporated a new in-processing method called "Reg[3]" as our baseline. The core idea behind this baseline is to add the fairness objective as regularization. To validate our approach, we conducted additional experiments on the MovieLens-1M dataset, utilizing two user models (PMF and NCF). The results（i.e.,Table 1, Table 2） demonstrate that our methods (FairLISA and FairGo+FairLISA) outperform other techniques, affirming the efficacy of our research.
>
> Table 1: The fairness and accuracy performance on PMF. The AUC represents AUC
> scores of all attackers. The smaller values of AUC denote better fairness performance with less
> sensitive information leakage (the fairer). GEN, AGE, OCC represent gender, age, and occupation. RMSE
> represents accuracy performance.
> |   Baseline   | GEN AUC | AGE AUC | OCC AUC | RMSE |
> | --------------- | ---------- | ------- | ------- | ------- |
> | Reg | 0.6483 | 0.7368 | 0.6631 | 0.8945 |
> |FairLISA | 0.5174 | **0.5276** | 0.5110 | 0.8912 |
> | FairGo+FairLISA | **0.5147** | 0.5316 | **0.5103** | **0.8812** |
>
> Table 2: The fairness and accuracy performance on NCF. The detail of evaluation metrics is the same as Table 1.
>
> |      Baseline   | GENDER AUC | AGE AUC | OCC AUC | RMSE|
> | --------------- | ---------- | ------- | ------- |------- |
> | Reg |    0.6814 | 0.5722 | 0.5886 | 0.8944 |
> |FairLISA | 0.5301 | 0.5218 | 0.5219 | **0.8831** |
> | FairGo+FairLISA | **0.5217** | **0.5205** | **0.5200** | 0.8892 |
>
> >**Q3. This paper only empirically studies the effectiveness of the proposed method. It is unknown whether or not the proposed method is theoretically better than the previous estimation-based method. It is suggested the authors could add some theoretical analysis to demonstrate that the proposed method is better than the previous estimation method.**
>
> A3: Thank you for your insightful suggestion. We acknowledge that conducting a theoretical analysis to establish the superiority of the proposed method is undeniably crucial. Nonetheless, it's imperative to recognize that addressing this challenge presents a significant undertaking. As an alternative avenue for evaluation,  we chose to substantiate our superiority through experiential insights. We will regard the theoretical substantiation of FairLISA as future work. Thanks again for your valuable suggestion.
>
> >**Q4. The authors are encouraged to provide code for reproduction.**
>
> A4: We greatly appreciate your suggestions. We strongly support open-source principles and will make the code public if our paper is fortunate enough to be accepted.
>
> >**Q5. Related references.**
>
> A5: Thanks for providing the references. These works are indeed pertinent to our research. We will incorporate these relevant references into the revision.
>
> [1] Yunqi Li, Hanxiong Chen, Shuyuan Xu, Yingqiang Ge, and Yongfeng Zhang. Towards personalized fairness based on causal notion. In Proceedings of the 44th International ACM SIGIR Conference on Research and Development in Information Retrieval, 2021.
>
> [2] Le Wu, Lei Chen, Pengyang Shao, Richang Hong, Xiting Wang, and Meng Wang. Learning fair
> representations for recommendation: A graph-based perspective. In Proceedings of the Web Conference 2021, 2021
>
> [3]Yao S, Huang B. Beyond parity: Fairness objectives for collaborative filtering[J]. Advances in neural information processing systems, 2017, 30.

---

> > ### Comment · Reviewer_teqm · 2023-08-18
> > **Thank you for the response.**
> >
> > Thank you for the response. There is no additional comment currently.

---

### Official Review · Reviewer_KpgW · 2023-07-23

**Soundness:** 2 fair
**Presentation:** 2 fair
**Contribution:** 2 fair
**Rating:** 4
**Confidence:** 4

**Summary:**

The paper proposed an algorithm to train fair user modeling (e.g recommender systems) when given limited sensitive attributes. The idea is to factorize out the effect of sensitive attributes from the model's fair training objectives, and isolate its impact in training.

**Strengths:**

1. Studied an important problem in practical fair recommender system

**Weaknesses:**

1. The proposed algorithm only applies to generative-based user modeling, which is limited. Many user modeling methods are prediction-based.
2. The high-level idea of decomposing the mutual information between prediction/embedding and sensitive attribute into sensitive-attribute-related term and non-sensitive-attribute-related term is common in the fairness literature. It seems limited novelty in the method.
3. It seems the method still requires some samples with sensitive attributes labeled (the minimum in experiments is 20%). It would be good to clarify how practitioners can obtain those sensitive attributes. Because if the concern is about privacy, then getting 20% samples with sensitive attributes is as hard as getting 100%. In the literature, usually, the assumption is having some aggregated form of sensitive attributes rather than sample-level.
4. A simple baseline is to train a classifier on samples with sensitive attributes and use it to label other data. This should be tested in experiments.
5. Many grammatical errors and typos.

**Questions:**

1. The evaluation focuses on privacy, which is how accurately an attacker can predict sensitive attributes from the user representations. But this evaluation would assume attackers can access user representation, which means the attacker is the model trainer, e.g. recommender system employer. Is it the case? If so, how can the attacker obtain the ground-truth sensitive attributes as the training labels of the attack model?

**Limitations:**

See Weaknesses.

---

> ### Author Rebuttal · Authors · 2023-08-09
>
> >**Q1. The proposed algorithm only applies to generative-based user modeling, which is limited. Many user modeling methods are prediction-based.**
>
> A1. Sorry for the confusion! I quite agree that currently many user modeling methods are prediction-based.  In fact,  I'd like to clarify that the user models discussed in this paper are prediction-based models that do not have generative capabilities [5][6]. Our FairLISA is specifically used to ensure fairness guarantees for these prediction-based models, which borrows the idea of GAN to remove the effect of sensitive attributes, as shown in Section 4 of the paper. In the revised version,  we would reorganize this part to make it clearer. In the future, we intend to expand our work to generative-based user modeling, such as diffusion-based user models [7] and LLM-based user models [8].
>
> >**Q2. The high-level idea of decomposing the mutual information between prediction/embedding and sensitive attribute into sensitive-attribute-related term and non-sensitive-attribute-related term is common. It seems limited novelty in the method.**
>
> A2. In this paper, we study fairness in situations with limited sensitive attributes. Our core contribution lies in efficiently leveraging data with both known and unknown sensitive attributes to facilitate fair model training. To achieve this, we design a theoretical-driven model.  Different from the conventional mutual information-based framework, our framework stands out in two significant aspects: (1) We employ distinct approaches for handling two types of data. (2) We provide theoretical guarantees for both types of data. Moreover, the experimental results further validate the SOTA performance achieved by our approach. Thanks for your question, we would reorganize our paper to make it clearer.
> >**Q3. It would be good to clarify how practitioners can obtain those sensitive attributes. Because if the concern is about privacy, then getting 20% samples with sensitive attributes is as hard as getting 100%. In the literature, usually, the assumption is having some aggregated form of sensitive attributes rather than sample-level.**
>
> A3. I sincerely appreciate your suggestion. In fact, the fairness setting with certain sample-level sensitive attributes available has been proposed in the literature, such as [1][2][3].  The limited sensitive attribute labels can be obtained through various methods. Firstly, on open platforms, certain users may publicly share their profiles, like the example of 14% of teen users who expose their complete profiles on Meta [4]. Secondly, some researchers rely on human annotators to label limited sensitive attributes [3]. We will incorporate your suggestions in the revision of our paper. Moreover, we will explore methods to achieve fairness in sensitive properties using aggregate forms.
>
> >**Q4. A simple baseline is to train a classifier on samples with sensitive attributes and use it to label other data. This should be tested in experiments.**
>
> A4. We have already compared this baseline in our paper, called FairGNN. The core idea behind FairGNN is to train an estimator to predict missing sensitive labels using data with known sensitive attributes (as described in the Introduction, Section 6.1, and Appendix C.1). Furthermore, we discussed FairGNN in detail in Section 5. The experimental results demonstrate the effectiveness of our method when compared to the baseline. We will reorganize our paper in the reversion to remove any lingering doubts or confusion.
>
> >**Q5. Many grammatical errors and typos.**
>
> A5. Thank you for pointing out grammatical errors and typos. We will revise our paper to fix these errors.
>
> >**Q6. How can the attacker obtain the ground-truth sensitive attributes as the training labels of the attack model in the evaluation?**
>
> A6. Sorry for the confusion. The attacker is indeed a recommender system employer. Nevertheless, during the testing phase, we are assuming that the labels for sensitive attributes are obtainable. These labels serve solely as a ground truth to assess the attacker's ability to predict sensitive attributes, which in turn allows us to access the fairness of our model. Note that during the training phase, only a subset of sensitive labels are accessible, aligning with our real-world scenario. Such a setting is commonly adopted in situations involving limited sensitive attributes [1][2][3]. In the future, we will make the necessary revisions to elucidate these aspects in our paper and address your concerns.
>
> [1]Zhang F, Kuang K, Chen L, et al. Fairness-aware contrastive learning with partially annotated sensitive attributes[C]//The Eleventh International Conference on Learning Representations. 2022.
>
> [2]Kırnap Ö, Diaz F, Biega A, et al. Estimation of fair ranking metrics with incomplete judgments[C]//Proceedings of the Web Conference 2021. 2021: 1065-1075.
>
> [3]Enyan Dai and Suhang Wang. Say no to the discrimination: Learning fair graph neural networks with limited sensitive attribute information. In Proceedings of the 14th ACM International Conference on Web Search and Data Mining, 2021.
>
> [4]Mary Madden, Amanda Lenhart, Sandra Cortesi, Urs Gasser, Maeve Duggan, Aaron Smith,
> 486 and Meredith Beaton. Teens, social media, and privacy. Pew Research Center, 2013.
>
> [5]Xiangnan He, Lizi Liao, Hanwang Zhang, Liqiang Nie, Xia Hu, and Tat-Seng Chua. Neural
> collaborative filtering. In Proceedings of the 26th international conference on world wide web, 2017.
>
> [6]Fei Wang, Qi Liu, Enhong Chen, Zhenya Huang, Yuying Chen, Yu Yin, Zai Huang, and Shijin
> Wang. Neural cognitive diagnosis for intelligent education systems. In Proceedings of the AAAI Conference on Artificial Intelligence, 2020.
>
> [7]Wang W, Xu Y, Feng F, et al. Diffusion Recommender Model[J]. arXiv preprint arXiv:2304.04971, 2023.
>
> [8]Wu L, Zheng Z, Qiu Z, et al. A Survey on Large Language Models for Recommendation[J]. arXiv preprint arXiv:2305.19860, 2023.

---

### Author Rebuttal · Authors · 2023-08-09

We sincerely appreciate all reviewers' time and efforts in reviewing our paper. We would like to thank all of them for providing constructive and valuable feedback which we will leverage to improve this work. Meanwhile, we are encouraged by the positive comments from reviewers, including:

**Motivation:** "Studied an important problem in practical fair recommender system" (Reviewer KpgW),  "an important and practical problem" (Reviewer teqm), "Studying limited sensitive attribute scenario is practical and important."(Reviewer Gbnv), "a socially valuable problem" (Reviewer xseN), "both meaningful and worthy of investigation" (Reviewer QZCP)

**Theoretical Contribution:** "a theoretical-driven perspective" (Reviewer teqm), "supported by the theoretical analysis" (Reviewer Gbnv), "with theoretical guarantees" (Reviewer xseN), "providing a simple yet effective approach, supported by theoretical guarantees." (Reviewer QZCP)

**Method:** "a novel adversarial learning method" (Reviewer teqm), " reasonable solutions" (Reviewer xseN), " straightforward and effective" (Reviewer xseN), "a look-nice model design"(Reviewer xseN), "a simple yet effective approach" (Reviewer QZCP)

**Experimental Results:** "Solid experiments are conducted" (Reviewer teqm), "performance is good based on the empirical evaluation." (Reviewer Gbnv), "a comprehensive validation experiments."(Reviewer xseN), "demonstrate the effectiveness and robustness of FairLISA in two representative user modeling tasks." (Reviewer QZCP)

We have provided specific responses to each reviewer. We hope our responses can clarify all your confusion and alleviate all concerns. We thank all reviewers again. Looking forward to your reply!

---

### Decision · Program_Chairs · 2023-09-21

**Decision:**

Accept (poster)

**Comment:**

Fairness with limited sensitive attribute information is a practically important and relevant setting. The proposed FairLISA framework applies the idea of using unlabeled data to perform fair training. The solution reduces the requirement of providing sensitive information. The reviewers agreed on the rigorousness and soundness of the results. The authors are encouraged to incorporate the discussions incurred during rebuttal in the final version. The camera-ready version can benefit from better positioning the contributions in light of the literature of fair training without sensitive attributes or with only limited and noisy sensitive attributes.